# Repurposing Pretrained Models for Robust Out-of-domain Few-Shot Learning

**Namyeong Kwon, Hwidong Na**[*]
Samsung Advanced Institute of Technology (SAIT), South Korea
{ny.kwon,hwidong.na}@samsung.com

**Gabriel Huang**
Mila, Université de Montréal

**Simon Lacoste-Julien**[†]
Mila, Université de Montréal
SAIT AI Lab, Montreal

## Abstract

Model-agnostic meta-learning (MAML) is a popular method for few-shot learning but assumes that we have access to the meta-training set. In practice, training on the meta-training set may not always be an option due to data privacy concerns, intellectual property issues, or merely lack of computing resources. In this paper, we consider the novel problem of repurposing pretrained MAML checkpoints to solve new few-shot classification tasks. Because of the potential distribution mismatch, the original MAML steps may no longer be optimal. Therefore we propose an alternative meta-testing procedure and combine MAML gradient steps with adversarial training and uncertainty-based stepsize adaptation. Our method outperforms "vanilla" MAML on same-domain and cross-domains benchmarks using both SGD and Adam optimizers and shows improved robustness to the choice of base stepsize.

## 1 Introduction

Deep learning approaches have shown improvements based on massive datasets and enormous computing resources. Despite their success, it is still challenging to apply state-of-the-art methods in the real world. For example, in semiconductor manufacturing (Nishi & Doering, 2000), collecting each new data point is costly and time consuming because it requires setting up a new manufacturing process accordingly. Moreover, in the case of a "destructive inspection", the cost is very high because the wafer must be destroyed for measurement. Therefore, learning from small amounts of data is important for practical purposes.

Meta-learning (learning-to-learn) approaches have been proposed for learning under limited data constraints. A meta-learning model optimizes its parameters for the best performance on the distribution of tasks. In particular, few-shot learning (FSL) formulates "learning from limited data" as an $n$-way $k$-shot problem, where $n$ is the number of classes and $k$ is the number of labeled samples per class. For each task in FSL, a support set is provided for training, while a query set is provided for evaluation. Ideally, a meta-learning model trained over a set of tasks (meta-training) will generalize well to new tasks (meta-testing).

Model-agnostic meta-learning (MAML) (Finn et al., 2017) is a general end-to-end approach for solving few-shot learning tasks. MAML is trained on the meta-training tasks to learn a model initialization (also known as *checkpoint*) such that a few gradient steps on the support set will yield the best predictions on the query set.

However, in practice it may not always be possible to retrain or finetune on the meta-training set. This situation may arise when the meta-training data is confidential, subject to restrictive licences, contains private user information, or protected intellectual property such as semiconductor manu-

---

[*]Work done as visiting researchers at SAIT AI Lab, Montreal.
[†]Canada CIFAR AI Chair

facturing know-how. Another reason is that one may not have the computing resources necessary for running large-scale meta-training.

In this paper, we consider the novel problem of *repurposing* MAML checkpoints to solve new few-shot classification tasks, *without* the option of (re)training on the meta-training set. Since the meta-testing set (new tasks) may differ in distribution from the meta-training set, the implicit assumption –in end-to-end learning– of identically distributed tasks may not hold, so there is no reason why the meta-testing gradient steps should match the meta-training. Therefore, we investigate various improvements over the default MAML gradient steps for test time adaptation.

Conceptually, our approach consists of collecting information while training the model on a new support set and then proposing ways to use this information to improve the adaptation. In this paper, we consider the variance of model parameters during ensemble training as a source of information to use. We propose algorithms that uses this information both to *adapt* the stepsizes for MAML as well as to generate "task-specific" adversarial examples to help robust adaptation to the new task.

Our **main contributions** are the following:

- We motivate the novel problem of *repurposing* MAML checkpoints to solve cross-domain few-shot classification tasks, in the case where the meta-training set is *inaccessible*, and propose a method based on uncertainty-based stepsize adaptation and adversarial data augmentation, which has the particularity that meta-testing differs from meta-training steps.
- Compared to "vanilla" MAML, our method shows improved accuracy and robustness to the choice of base stepsizes on popular cross-domain and same-domain benchmarks, using both the SGD and Adam (Kingma & Ba, 2014) optimizers. Our results also indicate that adversarial training (AT) is helpful in improving the model performance during meta-testing.

To the best of our knowledge, our work is the first few-shot learning method to combine the use of ensemble methods for stepsize computation and generating adversarial examples from the meta-testing support set for improved robustness. Moreover, our empirical observation of improving over the default meta-testing procedure of MAML motivates further research on alternative ways to leverage published model checkpoints.

## 2 RELATED WORK

### 2.1 META-LEARNING AND MODEL-AGNOSTIC META-LEARNING

FSL approaches deal with an extremely small amount of training data and can be classified into three categories. First, metric-based approaches solve few-shot tasks by training a feature extractor that maximizes inter-class similarity and intra-class dissimilarity (Vinyals et al., 2016; Snell et al., 2017; Sung et al., 2018). Second, memory-based approaches utilize previous tasks for new tasks with external memories (Santoro et al., 2016; Mishra et al., 2018). Third, optimization-based approaches search for good initial parameters during training and adapt the pretrained model for new tasks (Finn et al., 2017; Lee & Choi, 2018; Grant et al., 2018). We focus on the optimization-based approaches and suggest better training especially for the general family of MAML methods.

### 2.2 UNCERTAINTY FOR MODEL TRAINING

Uncertainty is an important criterion for measuring the robustness of a neural network (NN). Bayesian neural networks (BNN) (Blundell et al., 2015) obtain model uncertainty by placing prior distributions over the weights $p(\omega)$. This uncertainty has been used to adapt the stepsizes during continual learning in Uncertainty-guided Continual BNNs (UCB) (Ebrahimi et al., 2020). For each parameter, UCB scales its stepsize inversely proportional to the uncertainty of the parameter in the BNN to reduce changes in important parameters while allowing less important parameters to be modified faster in favor of learning new tasks. Our approach also decreases the stepsizes for "uncertain" parameters, but using a different notion of uncertainty, and instead in the context of FSL with a pretrained MAML checkpoint.

Unlike BNNs, deep ensembles (Lakshminarayanan et al., 2017) estimate uncertainty of NNs using ensembles over randomly initialized parameters combined with adversarial training (AT). Deep ensembles have been successfully used to boost predictive performance and AT has been used to im-

prove robustness to adversarial examples. Lakshminarayanan et al. (2017) showed that they produce predictive uncertainty estimates comparable in quality to BNNs. Deep ensembles, however, are not directly applicable for FSL tasks, as training randomly initialized parameters from scratch with a limited amount of training data yields poor performance. Our approach is partly inspired from deep ensembles, adapting it to the FSL setting by using instead a parameter perturbations of the MAML checkpoint model rather than a random initialization. We use in particular a multiplicative Gaussian perturbation that rescales the parameters, as the information content of the weights is said to be invariant to their scale (Wen et al., 2018).

### 2.3 ADVERSARIAL TRAINING FOR ATTACK AND DEFENSE

Deep NNs are sensitive to adversarial attacks (Goodfellow et al., 2015; Madry et al., 2018; Moosavi-Dezfooli et al., 2016). These methods generate an adversarial sample to fool a trained model, where the generated image looks identical to the original one for humans. Goodfellow et al. (2015) proposed the fast gradient sign method (FGSM) which generates adversarial example using sign of input gradient. While stronger attacks have been proposed (such as using projected gradient descent (Madry et al., 2018)), we will focus on the FGSM approach in this paper for simplicity. In FSL, adversarial attack and defense have also been studied. ADversarialMeta-Learner (Yin et al., 2018) utilized one-step adversarial attack for generating adversarial samples during meta-training. There is little consideration, however, for the degradation of accuracy in the original sample in adversarial defense approaches. Xie et al. (2020) reported improvement using generated examples for adversarial attack on the large scale training. Motiian et al. (2017) propose adversarial domain adaptation for FSL. Despite similar keywords, their work is distinct from ours and relies on using generative adversarial networks *discriminators* to confuse domains, whereas we rely on adversarial *examples* for improved robustness. To our best knowledge, there is no prior FSL work which uses adversarial examples to enhance the model at test-time.

### 2.4 DOMAIN ADAPTATION

Domain adaptation methods (DA) (Ben-David et al., 2010) attempt to alleviate the distribution mismatch between source and target domains. Most of the recently proposed DA approaches are based on generative adversarial networks (GANs) (Goodfellow et al., 2014). GAN-based DA approaches require having access to the large amount of unlabeled data from both the source and target datasets (Tzeng et al., 2017; Zhang et al., 2019; Wilson & Cook, 2020). DA methods cannot be directly applied in the FSL scenario due to limited number of target domain samples (*shots*). Some domain-adaptive FSL (DA-FSL) methods have been proposed in the case where only a very few samples from the target domain are available (Motiian et al., 2017; Zhao et al., 2020).

DA and DA-FSL methods cannot be directly applied to our setting because we assume that the meta-training dataset is *inaccessible* — mirroring real world situations in which access to the meta-training set is restricted by privacy and confidentiality concerns. Our approach differs from DA and DA-FSL by not requiring access to the meta-training dataset (source domain).

## 3 PROPOSED METHOD

At meta-testing time, MAML normally uses the support set to compute *fixed* gradient steps, which were "calibrated" using end-to-end learning during meta-training. That is, the learned initialization is such that a fixed combination of *stepsize* and *loss* result in the desired result. However, those stepsizes and losses may be suboptimal on the new task, especially if the new task is out-of-domain with respect to the meta-training tasks.

Our method is based on the assumption that the support set can be used to improve the meta-testing procedure itself, beyond merely serving as training examples. We start by leveraging the support set to estimate task-specific uncertainties over the model parameters. Then, we propose two improvements over the "vanilla" MAML gradient steps : we *scale* the gradient steps using layer-wise stepsizes computed from the support set and we train using *task-specific adversarial examples*.

### 3.1 MOTIVATING HYPOTHESES

At meta-testing time, we start by assuming that we can estimate *task-specific uncertainties* over the model parameters. One possibility, which we adopt in Section 3.2, is to train deep ensembles (Lakshminarayanan et al., 2017) on the support set and use the ensemble to estimate variances over model parameters. Each model learns slightly different parameters, and yields slightly different gradients. We regard the variance over the parameters and input gradients as task-specific uncertainties. Given the uncertainty estimates, we propose two modifications to the original MAML gradient steps.

> **Proposal 1 (Task-specific stepsizes)** *Use lower stepsizes for model parameters with higher variance.*
>
> *In our case, the variance could be further amplified if high-variance components were to be moved with large stepsizes. Therefore, we carefully move high-variance components with a lower stepsize with the hope is that we can limit the variance over the model parameters. This approach can be related with the fact that lower stepsizes should be taken for SGD when the gradients are very noisy (Dieuleveut et al., 2020).*

> **Proposal 2 (Task-specific adversarial examples)** *Use adversarial examples with higher adversarial perturbation on input components with higher variance over the input gradient.*
>
> *The intuition is that if only slightly perturbed models (from the ensemble) disagree on parts of the input gradient, then it means that they disagree on what to learn and therefore those parts of the input are more vulnerable to attack. Therefore, we propose to use AT with stronger adversarial perturbation in the weak parts of the input, with the hope to incur improved robustness on those parts of the input.*

We regard adversarial training as a form of *data augmentation* or *regularization* at meta-testing time, which we hope allows the model to overcome the limited size of the support set.

### 3.2 UNCERTAINTY-BASED GRADIENT STEPS AT TEST-TIME

We improve over the default MAML gradient steps by implementing the ideas presented in the previous section. The resulting approach is detailed in **Algorithm 1** and is a combination of uncertainty-based stepsize adaption (USA, based on Proposal 1) and uncertainty-based fast gradient sign method (UFGSM), adversarial training (AT), and generating additional adversarial examples from deep ensembles (EnAug) which are based on Proposal 2.

Denote $\mathcal{L}(D, \theta) = 1/|D| \sum_{(x,y) \in D} l_\theta(x, y)$ the cross-entropy loss for model $\theta$ over the labeled dataset $D = \{(x, y) \dots\}$, $\mathcal{A}_\theta(x, y) = x + \epsilon \mathtt{sign}(\nabla_x l_\theta(x, y))$ the FGSM adversarial example computed from $(x, y)$ and $\mathcal{L}_{\mathrm{AT}}(D, \theta) = 1/|D| \sum_{(x,y) \in D} l_\theta(\mathcal{A}_\theta(x, y), y)$ the resulting adversarial cross-entropy, which we refer to as **AT**. For easy reference, all notations are summarized in **Appendix A.1**.

Starting from the pretrained MAML checkpoint $\theta_0$, we perturb the model parameters with multiplicative Gaussian noise to create a deep ensemble $(\theta_0^m)_{1 \leq m \leq M}$ (**lines 2-4**). Then, we repeat the following for $T$ steps. At time $t$, run gradient descent on each model $\theta_t^m$ of the ensemble (**line 7**), where the *loss* is a combination of the usual cross-entropy $\mathcal{L}$ and AT loss $\mathcal{L}_{\mathrm{AT}}$ as in (Lakshminarayanan et al., 2017) but on the support set, and the stepsizes $\alpha^{\mathrm{adap}}$ are updated using **USA** (details below). Also, we generate adversarial examples using FGSM and **UFGSM** (details below) and store them into $D^{\mathrm{Aug}}$ (**lines 8 and 11**). Finally, we run $T$ gradient steps on the original checkpoint, where the loss is a combination of cross-entropy on the support set $D^{\mathrm{Spt}}$, AT loss on the support set, and cross-entropy on the ensemble-augmented support set $D^{\mathrm{Aug}}$ (**lines 13-16**). Note that we recover the original MAML steps for $(\lambda_{\mathrm{AT}} = \lambda_{\mathrm{Aug}} = 0, \lambda_\alpha = 1)$, while we refer to the case $(\lambda_{\mathrm{AT}} = \lambda_{\mathrm{Aug}} = 1, \lambda_\alpha = 0)$ as our full method.

---

**Algorithm 1:** Uncertainty-based Gradient Steps at Test-time

---

**Data:** New task support set $D^{\text{Spt}} = \{(x_1, y_1), \ldots, (x_{n \times k}, y_{n \times k})\}$ with $n$ ways and $k$ shots.
**Require:** Base stepsize $\alpha$, pretrained weights $\theta_0$, Gaussian variance $\sigma$, AT coefficient $\epsilon$, size of ensemble $M$, number of gradient steps $T$, selection coefficients in $\{0, 1\}$ for: base stepsize $\lambda_\alpha$, adversarial loss $\lambda_{\text{AT}}$, and augmented cross-entropy $\lambda_{\text{Aug}}$.

1   $D^{\text{Aug}} \leftarrow \emptyset, \alpha^{\text{adap}} \leftarrow \alpha$

2   **for** $m = 1$ **to** $M$ **do**
3      $\theta_0^m \leftarrow \theta_0(1 + \mathcal{N}(0, \sigma))$                   ▷Initialize deep ensemble
4   **end**

5   **for** $t = 1$ **to** $T$ **do**
6      **for** $m = 1$ **to** $M$ **do**
7          $\theta_t^m \leftarrow \theta_{t-1}^m - \alpha^{\text{adap}} \odot \nabla_{\theta_{t-1}^m}[\mathcal{L}(D^{\text{Spt}}, \theta_{t-1}^m) + \mathcal{L}_{\text{AT}}(D^{\text{Spt}}, \theta_{t-1}^m)]$
8          $D^{\text{Aug}} \leftarrow D^{\text{Aug}} \cup \{(\mathcal{A}_{\theta_{t-1}^m}(x, y), y) \,|\, (x, y) \in D^{\text{Spt}}\}$          ▷EnAug
9      **end**
10     $\alpha^{\text{adap}} \leftarrow \text{USA}(\alpha, \theta_t^{1:M})$                           ▷Algorithm 2
11     $D^{\text{Aug}} \leftarrow D^{\text{Aug}} \cup \{\, \text{UFGSM}(\theta_0, \theta_t^{1:M}, x, y) \,|\, (x, y) \in D^{\text{Spt}}\}$      ▷Algorithm 3
12   **end**
13   $\alpha \leftarrow \lambda_\alpha \alpha + (1 - \lambda_\alpha)\alpha^{\text{adap}}$
14   **for** $t = 1$ **to** $T$ **do**
15      $\theta_t \leftarrow \theta_{t-1} - \alpha \odot \nabla_{\theta_{t-1}}[\mathcal{L}(D^{\text{Spt}}, \theta_{t-1}) + \lambda_{\text{AT}}\mathcal{L}_{\text{AT}}(D^{\text{Spt}}, \theta_{t-1}) + \lambda_{\text{Aug}}\mathcal{L}(D^{\text{Aug}}, \theta_{t-1})]$
16   **end**

---

**USA.** We propose *uncertainty-based stepsize adaptation* (USA), which assigns lower stepsizes to layers[1] with higher uncertainty (**Algorithm 2**) – we call this loosely an "inverse-relationship" below. Let $\alpha$ denote the default (scalar) stepsize and $\theta^{1:M}$ the parameters of the ensemble models, where $M$ is the size of the ensemble and the number of layers in model is $L$. To compute the adapted layer-wise stepsizes $\alpha^{\text{adap}}$, we compute $u$, the parameter-wise standard deviation of the model parameters over the ensemble (**line 2**), apply an inverse-relationship transformation which flips the max with the min (**line 3**), average over each layer (**line 4**) and $L_1$-normalize the result (**line 5**). The design choices for USA is explained in **Appendix A.2**. The resulting stepsizes $\alpha^{\text{adap}}$ have an inverse-relationship with the variance of each layer. We give an example of applying USA to the 4-ConvNet architecture on *mini*ImageNet (**Figure 1**). On the left, we plot the layer-wise standard deviations of the parameters and on the right the corresponding USA stepsizes, which follow an inverse relationship with the standard deviations.

---

**Algorithm 2:** USA

---

1 **Function** $\text{USA}(\alpha, \theta_t^{1:M})$
2      $u = \text{Std}(\theta_t^{1:M})$
3      $c = \text{Max}(u) - u + \text{Min}(u)$
4      $\mu^l \leftarrow$ average of $c$ over each layer $l$
5      $\alpha^{\text{adap}} = \alpha\mu^l/(1/L \sum_{l=1}^{L} \mu^l)$
6      **return** $\alpha^{\text{adap}}$
7 **end**

---

**Algorithm 3:** UFGSM

---

**Define:** $\text{MinMaxNorm}(x) = \frac{x - \text{Min}(x)}{\text{Max}(x) - \text{Min}(x)}$
1 **Function** $\text{UFGSM}(\theta, \theta_t^{1:M}, x, y)$
2      $u = \text{Std}(\{\nabla_x l_{\theta'}(x, y) \,|\, \theta' \in \theta_t^{1:M}\})$
3      $u = \text{MinMaxNorm}(u)$
4      $x' = x + \epsilon u \odot \text{sign}(\nabla_x l_\theta(x, y))$
5      **return** $x'$
6 **end**

---

**UFGSM.** We also propose *uncertainty-based FGSM* (UFGSM) to generate adversarial examples, with higher adversarial perturbation on input components with higher variance over the input gradient (**Algorithm 3**). Starting from an image $x$ and label $y$, we compute the input gradient for each model from the ensemble and calculate the standard deviation $u$ over the ensemble (**line 2**). Then, we linearly map $u$ between 0 and 1 (**line 3**), compute the FGSM adversarial example for the pre-

---

[1]In this study, we choose to use layer-wise stepsizes, but it is also possible to use kernel-wise or parameter-wise stepsizes. The motivation for using layer-wise stepsizes is because features from the same layer tend to have the same level of abstraction (Zeiler & Fergus, 2014).

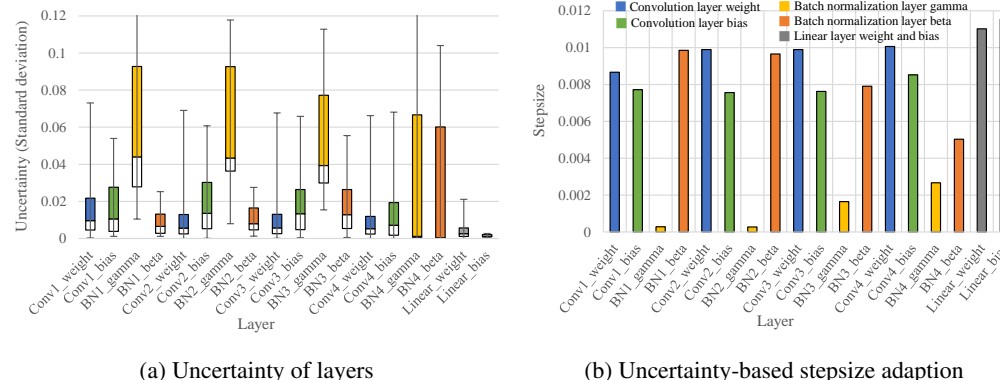

(a) Uncertainty of layers         (b) Uncertainty-based stepsize adaption

Figure 1: USA converts uncertainty into layer-wise adapted stepsize. Here $\alpha = 0.01$. (a) is standard deviation of trained ensemble models' weights. Each layer has a difference uncertainty. (b) is adapted stepsize by USA. Each layer has a different stepsize based on the uncertainty.

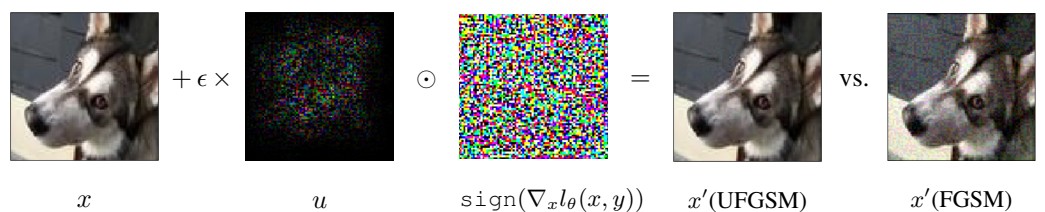

$x$          $u$      $\text{sign}(\nabla_x l_\theta(x,y))$     $x'(\text{UFGSM})$     $x'(\text{FGSM})$

Figure 2: Applying UFGSM to 4-ConvNet on *mini*ImageNet with $\epsilon = 0.05$. Starting from the clean image $x$, we add the signed gradient $\text{sign}(\nabla_x l_\theta(x,y))$ after rescaling it by the uncertainty over the input gradient $u$, to generate the adversarial example $x'(\text{UFGSM})$. Note how UFGSM generated a more natural image than FGSM (rightmost, $u = 1$).

trained model $\theta_0$ and rescale it using $u$, so that areas of higher variance get more perturbation. We give an example of applying UFGSM to *mini*ImageNet in **Figure 2**.

## 4 EXPERIMENTAL EVALUATION

We train MAML on *mini*ImageNet (Vinyals et al., 2016) training split; we then apply our method on the resulting checkpoint. We evaluate our model on the test split of *mini*ImageNet – for the same-domain setting – as well as CUB-200-2011 (Welinder et al., 2010), Traffic Sign (Houben et al., 2013) and VGG Flower (Nilsback & Zisserman, 2008) – for the cross-domain setting. These datasets are denoted as Mini, Birds, Signs and Flowers respectively. A desirable feature for an optimizer is to maintain good performance in a broad range of stepsizes (Asi & Duchi, 2019). Therefore, we evaluate our approach not only at the optimal stepsize, but also over a broad range of base stepsizes[2] from $10^{-4}$ to 1. We evaluate the performance with three metrics: All, Top-1 and Top-40%.[3] If two methods have comparable Top-1 performance, but one has better Top-40% performance, then that method is more robust to the choice of base stepsize. Detailed experimental setup and pretrained model selection are included in **Appendix A.3**. Our code is available at `https://github.com/NamyeongK/USA_UFGSM/`.

### 4.1 MAIN RESULTS

Our main results are in **Table 1**. We compare the default MAML steps, which we take as our baseline (denoted as SGD), with our full method (denoted as SGD+All), which consists of combining

---

[2]We determine the ranges of stepsize based on training performance on *mini*ImageNet and keep it the same for other datasets. We selected the minimum and maximum stepsize where performance decreased drastically.

[3]All is the average accuracy over all stepsizes. Top-1 is the accuracy of the best performing stepsize. Top-40% is the average of the top 40% accuracies among all the stepsizes

Table 1: Main results. We compare the default MAML steps (SGD) with our method (SGD+All) on same-domain and cross-domain benchmarks.

| Metric | Dataset | 5-way 1-shot | | 5-way 5-shot | | 10-way 1-shot | |
|---|---|---|---|---|---|---|---|
| | | SGD | SGD+All | SGD | SGD+All | SGD | SGD+All |
| All | Mini | $\mathbf{37.21_{\pm 0.22}}$ | $36.24_{\pm 0.22}$ | $38.14_{\pm 0.39}$ | $\mathbf{43.48_{\pm 0.38}}$ | $21.31_{\pm 0.37}$ | $\mathbf{22.07_{\pm 0.4}}$ |
| | Birds | $\mathbf{30.07_{\pm 0.52}}$ | $29.83_{\pm 0.46}$ | $33.29_{\pm 0.29}$ | $\mathbf{38.07_{\pm 0.33}}$ | $17.54_{\pm 0.06}$ | $\mathbf{17.81_{\pm 0.12}}$ |
| | Flowers | $40.54_{\pm 0.48}$ | $\mathbf{40.86_{\pm 0.45}}$ | $44.55_{\pm 0.43}$ | $\mathbf{53.10_{\pm 0.55}}$ | $26.79_{\pm 0.41}$ | $\mathbf{28.07_{\pm 0.41}}$ |
| | Signs | $34.19_{\pm 0.76}$ | $\mathbf{34.76_{\pm 0.61}}$ | $45.39_{\pm 0.45}$ | $\mathbf{51.02_{\pm 0.5}}$ | $23.77_{\pm 0.13}$ | $\mathbf{24.88_{\pm 0.13}}$ |
| | Avg. | $\mathbf{35.50_{\pm 0.49}}$ | $35.42_{\pm 0.44}$ | $40.34_{\pm 0.39}$ | $\mathbf{46.42_{\pm 0.44}}$ | $22.35_{\pm 0.24}$ | $\mathbf{23.21_{\pm 0.26}}$ |
| Top-1 | Mini | $\mathbf{46.71_{\pm 0.81}}$ | $46.62_{\pm 0.58}$ | $\mathbf{62.46_{\pm 0.68}}$ | $62.16_{\pm 0.66}$ | $31.10_{\pm 0.66}$ | $\mathbf{31.42_{\pm 0.82}}$ |
| | Birds | $36.03_{\pm 0.48}$ | $\mathbf{36.43_{\pm 0.58}}$ | $50.71_{\pm 0.54}$ | $\mathbf{51.82_{\pm 0.64}}$ | $23.16_{\pm 0.21}$ | $\mathbf{23.61_{\pm 0.39}}$ |
| | Flowers | $51.97_{\pm 0.79}$ | $\mathbf{54.36_{\pm 1.04}}$ | $70.63_{\pm 0.76}$ | $\mathbf{74.52_{\pm 0.69}}$ | $38.61_{\pm 0.49}$ | $\mathbf{40.68_{\pm 0.58}}$ |
| | Signs | $43.44_{\pm 0.99}$ | $\mathbf{44.10_{\pm 1.03}}$ | $72.52_{\pm 0.84}$ | $\mathbf{74.13_{\pm 0.90}}$ | $35.10_{\pm 0.21}$ | $\mathbf{35.11_{\pm 0.26}}$ |
| | Avg. | $44.54_{\pm 0.77}$ | $\mathbf{45.37_{\pm 0.81}}$ | $64.08_{\pm 0.71}$ | $\mathbf{65.66_{\pm 0.72}}$ | $31.99_{\pm 0.39}$ | $\mathbf{32.70_{\pm 0.51}}$ |
| Top-40% | Mini | $45.07_{\pm 0.52}$ | $\mathbf{45.79_{\pm 0.44}}$ | $55.38_{\pm 0.46}$ | $\mathbf{60.70_{\pm 0.91}}$ | $28.65_{\pm 0.64}$ | $\mathbf{30.56_{\pm 0.77}}$ |
| | Birds | $34.88_{\pm 0.55}$ | $\mathbf{35.77_{\pm 0.55}}$ | $45.50_{\pm 0.49}$ | $\mathbf{51.03_{\pm 0.63}}$ | $21.78_{\pm 0.07}$ | $\mathbf{22.87_{\pm 0.30}}$ |
| | Flowers | $49.98_{\pm 0.69}$ | $\mathbf{53.16_{\pm 0.73}}$ | $65.16_{\pm 0.73}$ | $\mathbf{73.55_{\pm 0.86}}$ | $36.14_{\pm 0.53}$ | $\mathbf{39.75_{\pm 0.61}}$ |
| | Traffic | $41.81_{\pm 1.02}$ | $\mathbf{43.39_{\pm 1.01}}$ | $67.34_{\pm 0.43}$ | $\mathbf{71.45_{\pm 0.90}}$ | $33.56_{\pm 0.11}$ | $\mathbf{34.27_{\pm 0.17}}$ |
| | Avg. | $42.94_{\pm 0.70}$ | $\mathbf{44.52_{\pm 0.68}}$ | $58.34_{\pm 0.53}$ | $\mathbf{64.18_{\pm 0.82}}$ | $30.03_{\pm 0.34}$ | $\mathbf{31.86_{\pm 0.46}}$ |

USA, UFSGM, EnAug, and AT ($\lambda_{\text{AT}} = \lambda_{\text{Aug}} = 1, \lambda_{\alpha} = 0$ in **Algorithm 1**). In terms of absolute performance (Top-1 row), our method outperforms the baseline on cross-domain tasks (Birds, Flowers, Signs), while the performance is comparable on same-domain tasks (Mini). In terms of robustness to the base stepsize (All and Top-40% rows), our method outperforms the baseline over a large range of stepsizes for 5-way 5-shot and 10-way 1-shot tasks, while for 5-way 1-shot the results are either comparable (All) or better (Top-40%) depending on the metric considered. More results can be found in **Appendix A.6**.

## 4.2 DISCUSSION

**Ablation Study.** We perform an ablation study for the 5-way 1-shot case in **Table 2** and plot the accuracy at different base stepsizes for the Flowers dataset in **Figure 3**. Comparing SGD to SGD+AT in Table 2, we observe that adversarial training is beneficial over the baseline (Top1 and Top-40%), except for large stepsizes with SGD, which is reflected in the All metric and in the dip in performance in Figure 3a. Notice also how the use of AT flattens the accuracy curve near the optimal stepsize. Comparing SGD to SGD+USA, we observe a very small but consistent improvement over the baseline in the vicinity of the optimal stepsize (Top-1 and Top-40%). Comparing SGD+USA to SGD+USA+UFGSM, we observe improvement over a wide range of stepsizes, which is reflected in the curves and in All and Top-40% metrics. Comparing SGD+USA+UFGSM+EnAug+AT to SGD+USA+UFGSM+EnAug shows that AT+EnAug is beneficial in terms of absolute performance (Top-1) as well as creating robustness to the choice of stepsize (Top-40%). The drop for the All metric is explained by the dip in the curve for the largest stepsizes. Overall, the best absolute performance (Top-1) is always obtained through some use of *adversarial training*, while using the full method consistently results in *increased robustness* to the choice of stepsize (best Top-40% performance). **Appendix A.5** shows various AT results.

**UFGSM vs. FGSM.** We compare UFGSM to FGSM (Goodfellow et al., 2015) in **Table 3**. The results show that our uncertainty-based approach consistently outperforms FGSM, which suggests that the uncertainty information extracted from the support set was useful.

**SGD vs. Adam.** Our method can be used with SGD and Adam, however, we have mainly focused on SGD throughout the paper. SGD tends to yield the best results (Top-1 and Top-40%), as shown in **Table 4** for the baseline and full method. More results for Adam can be found in Sections A.6 and A.8 of the appendix.

**Best checkpoint vs. Overfitted checkpoint.** We also consider the problem of repurposing an overfitted checkpoint. We find that our method improves both absolute performance and robustness

Table 2: Ablation Study for 5-way 1-shot classification with SGD optimizer. SGD+AT corresponds to using ($\lambda_{\text{AT}} = \lambda_{\alpha} = 1, \lambda_{\text{Aug}} = 0$) in Algorithm 1, SGD+USA to ($\lambda_{\text{AT}} = \lambda_{\text{Aug}} = \lambda_{\alpha} = 0$), SGD+USA+UFGSM to ($\lambda_{\text{AT}} = \lambda_{\alpha} = 0, \lambda_{\text{Aug}} = 1$) and SGD+USA+UFGSM+EnAug+AT to ($\lambda_{\text{AT}} = \lambda_{\text{Aug}} = 1, \lambda_{\alpha} = 0$).

| Metric | Dataset | 5-way 1-shot | | | | |
| --- | --- | --- | --- | --- | --- | --- |
| | | SGD | SGD+AT | SGD+USA | SGD+USA +UFGSM | SGD+USA +UFGSM +EnAug+AT |
| All | Mini | $37.21_{\pm 0.22}$ | $34.60_{\pm 0.18}$ | $37.53_{\pm 0.27}$ | $\mathbf{38.71}_{\pm \mathbf{0.28}}$ | $36.24_{\pm 0.22}$ |
| | Birds | $30.07_{\pm 0.52}$ | $28.81_{\pm 0.45}$ | $30.39_{\pm 0.53}$ | $\mathbf{31.07}_{\pm \mathbf{0.50}}$ | $29.83_{\pm 0.46}$ |
| | Flowers | $40.54_{\pm 0.48}$ | $38.80_{\pm 0.44}$ | $41.09_{\pm 0.49}$ | $\mathbf{42.33}_{\pm \mathbf{0.60}}$ | $40.86_{\pm 0.45}$ |
| | Signs | $34.19_{\pm 0.76}$ | $33.42_{\pm 0.59}$ | $35.02_{\pm 0.82}$ | $\mathbf{36.10}_{\pm \mathbf{0.48}}$ | $34.76_{\pm 0.61}$ |
| | Avg. | $35.50_{\pm 0.49}$ | $33.91_{\pm 0.41}$ | $36.01_{\pm 0.36}$ | $\mathbf{37.05}_{\pm \mathbf{0.27}}$ | $35.42_{\pm 0.44}$ |
| Top-1 | Mini | $46.71_{\pm 0.81}$ | $46.58_{\pm 0.65}$ | $\mathbf{46.72}_{\pm \mathbf{0.71}}$ | $46.66_{\pm 0.77}$ | $46.62_{\pm 0.58}$ |
| | Birds | $36.03_{\pm 0.48}$ | $36.40_{\pm 0.48}$ | $36.08_{\pm 0.49}$ | $36.02_{\pm 0.48}$ | $\mathbf{36.43}_{\pm \mathbf{0.58}}$ |
| | Flowers | $51.97_{\pm 0.79}$ | $\mathbf{54.55}_{\pm \mathbf{0.84}}$ | $52.01_{\pm 0.77}$ | $52.30_{\pm 0.77}$ | $54.36_{\pm 1.04}$ |
| | Signs | $43.43_{\pm 0.99}$ | $43.98_{\pm 0.98}$ | $43.42_{\pm 1.06}$ | $43.96_{\pm 0.64}$ | $\mathbf{44.10}_{\pm \mathbf{1.03}}$ |
| | Avg. | $44.54_{\pm 0.77}$ | $\mathbf{45.38}_{\pm \mathbf{0.74}}$ | $44.56_{\pm 0.44}$ | $44.73_{\pm 0.41}$ | $45.37_{\pm 0.81}$ |
| Top-40% | Mini | $45.07_{\pm 0.52}$ | $45.53_{\pm 0.50}$ | $45.23_{\pm 0.59}$ | $45.76_{\pm 0.62}$ | $\mathbf{45.79}_{\pm \mathbf{0.44}}$ |
| | Birds | $34.88_{\pm 0.55}$ | $35.57_{\pm 0.60}$ | $35.17_{\pm 0.55}$ | $35.30_{\pm 0.48}$ | $\mathbf{35.77}_{\pm \mathbf{0.55}}$ |
| | Flowers | $49.98_{\pm 0.69}$ | $52.92_{\pm 0.68}$ | $50.39_{\pm 0.71}$ | $51.07_{\pm 0.67}$ | $\mathbf{53.16}_{\pm \mathbf{0.73}}$ |
| | Signs | $41.83_{\pm 1.02}$ | $43.19_{\pm 0.96}$ | $42.08_{\pm 1.06}$ | $43.18_{\pm 0.73}$ | $\mathbf{43.39}_{\pm \mathbf{1.01}}$ |
| | Avg. | $42.94_{\pm 0.70}$ | $44.30_{\pm 0.68}$ | $43.22_{\pm 0.43}$ | $43.83_{\pm 0.39}$ | $\mathbf{44.52}_{\pm \mathbf{0.68}}$ |

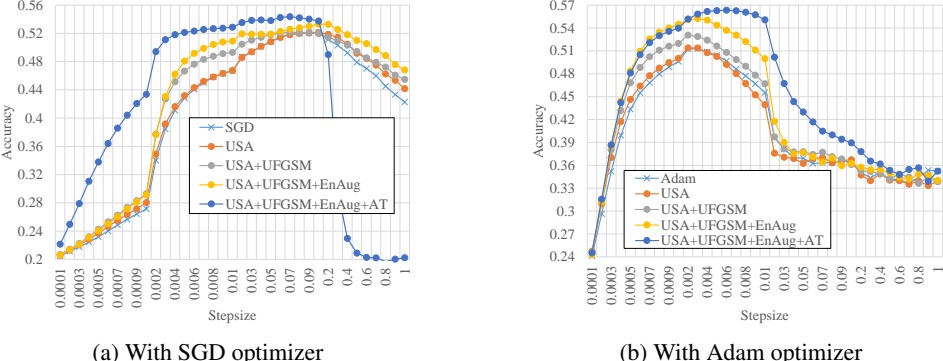

(a) With SGD optimizer      (b) With Adam optimizer

Figure 3: Ablation study for 5-way 1-shot classification on Flowers dataset.

Table 3: Comparing UFGSM against FGSM (Goodfellow et al., 2015) on 5-way 1-shot tasks. For all metrics and datasets, the proposed uncertainty-based method is better.

| Metric | Dataset | SGD+USA | |
|---|---|---|---|
| | | +UFGSM | +FGSM |
| All | Mini | $38.71_{\pm 0.28}$ | $36.97_{\pm 0.54}$ |
| | Birds | $31.07_{\pm 0.50}$ | $30.05_{\pm 0.55}$ |
| | Flowers | $42.33_{\pm 0.60}$ | $41.17_{\pm 0.55}$ |
| | Signs | $36.10_{\pm 0.48}$ | $34.78_{\pm 0.50}$ |
| | Avg. | $37.05_{\pm 0.27}$ | $35.74_{\pm 0.27}$ |
| Top-1 | Mini | $46.66_{\pm 0.77}$ | $43.93_{\pm 0.85}$ |
| | Birds | $36.02_{\pm 0.48}$ | $34.52_{\pm 0.74}$ |
| | Flowers | $52.30_{\pm 0.77}$ | $50.63_{\pm 0.84}$ |
| | Signs | $43.96_{\pm 0.64}$ | $42.62_{\pm 1.00}$ |
| | Avg. | $44.73_{\pm 0.41}$ | $42.93_{\pm 0.47}$ |
| Top-40% | Mini | $45.76_{\pm 0.62}$ | $43.18_{\pm 0.87}$ |
| | Birds | $35.30_{\pm 0.48}$ | $33.88_{\pm 0.75}$ |
| | Flowers | $51.07_{\pm 0.67}$ | $49.47_{\pm 0.69}$ |
| | Signs | $43.18_{\pm 0.73}$ | $41.92_{\pm 0.90}$ |
| | Avg. | $43.83_{\pm 0.39}$ | $42.11_{\pm 0.46}$ |

Table 4: Comparing SGD vs. Adam for default MAML steps (baseline) and our method (baseline+all) on 5-way 1-shot classification.

| Metric | Dataset | Baseline | | Baseline+All | |
|---|---|---|---|---|---|
| | | SGD | Adam | SGD | Adam |
| All | Mini | $37.21_{\pm 0.22}$ | $31.87_{\pm 0.20}$ | $36.24_{\pm 0.22}$ | $33.57_{\pm 0.29}$ |
| | Birds | $30.07_{\pm 0.52}$ | $28.02_{\pm 0.39}$ | $29.83_{\pm 0.46}$ | $29.51_{\pm 0.46}$ |
| | Flowers | $40.54_{\pm 0.48}$ | $40.33_{\pm 0.57}$ | $40.86_{\pm 0.45}$ | $44.58_{\pm 0.70}$ |
| | Signs | $34.19_{\pm 0.76}$ | $33.98_{\pm 0.43}$ | $34.76_{\pm 0.61}$ | $35.10_{\pm 0.43}$ |
| | Avg. | $35.50_{\pm 0.49}$ | $33.55_{\pm 0.22}$ | $35.42_{\pm 0.44}$ | $35.69_{\pm 0.29}$ |
| Top-1 | Mini | $46.71_{\pm 0.81}$ | $43.08_{\pm 0.10}$ | $46.62_{\pm 0.58}$ | $44.72_{\pm 0.35}$ |
| | Birds | $36.03_{\pm 0.48}$ | $34.00_{\pm 0.63}$ | $36.43_{\pm 0.58}$ | $35.45_{\pm 0.67}$ |
| | Flowers | $51.97_{\pm 0.79}$ | $51.28_{\pm 0.47}$ | $54.36_{\pm 1.04}$ | $56.04_{\pm 1.14}$ |
| | Signs | $43.43_{\pm 0.99}$ | $42.53_{\pm 0.79}$ | $44.10_{\pm 1.03}$ | $43.01_{\pm 0.85}$ |
| | Avg. | $44.54_{\pm 0.77}$ | $42.72_{\pm 0.32}$ | $45.37_{\pm 0.81}$ | $44.80_{\pm 0.54}$ |
| Top-40% | Mini | $45.07_{\pm 0.52}$ | $40.18_{\pm 0.14}$ | $45.79_{\pm 0.44}$ | $42.92_{\pm 0.35}$ |
| | Birds | $34.88_{\pm 0.55}$ | $32.38_{\pm 0.61}$ | $35.77_{\pm 0.55}$ | $34.54_{\pm 0.66}$ |
| | Flowers | $49.98_{\pm 0.69}$ | $48.18_{\pm 0.44}$ | $53.16_{\pm 0.73}$ | $54.08_{\pm 0.76}$ |
| | Signs | $41.83_{\pm 1.02}$ | $40.90_{\pm 0.70}$ | $43.39_{\pm 1.01}$ | $42.26_{\pm 0.77}$ |
| | Avg. | $42.94_{\pm 0.70}$ | $40.41_{\pm 0.29}$ | $44.52_{\pm 0.68}$ | $43.45_{\pm 0.40}$ |

to stepsize. In fact, the gains are more substantial on the overfitted (worse) checkpoint than the best checkpoint (**Appendix A.8**).

**Validating Proposal 1 and 2.** Our method is built on the hypotheses that we should take *small* stepsizes on *high* uncertainty parameters and add *more* adversarial perturbation on *high* uncertainty input gradients. To validate those choices, we have tried taking *large* stepsizes on *high* uncertainty parameters, and using *less* adversarial perturbation on *high* uncertainty input gradients, which resulted in lower accuracy and robustness to the stepsize (**Appendix A.7**).

**Freezing most uncertain layers.** We observed that the batch normalization (BN) layers have the most uncertainty (**Figure 1**). In Proposal 1, we argued that the higher uncertainty component amplifies the error when we use bigger stepsize. As suggested by an anonymous reviewer, this motivates an additional experiment that we perform where we "freeze" the BN layer during meta-testing (i.e. we manually set the stepsize for the BN scale and shift parameters to zero). We performed the BN layer freezing experiment on SGD+All (see detailed results in **Table 10** and **Figure 12** in **Appendix A.9**). It turns out that SGD+All (w/ freezing BN) outperforms SGD+All (w/o freezing BN) on the All metric, with most improvement in the higher stepsize range (>0.1). This is consistent with our intuition that updating high-uncertainty layers (such as BN) with large stepsizes can be harmful.

## 5 CONCLUSION

In this paper we considered the novel problem of repurposing pretrained MAML checkpoints for out-of-domain few-shot learning. Our method uses deep ensembles to estimate model parameter and input gradient uncertainties over the support set, and builds upon the default MAML gradient steps through the addition of uncertainty-based adversarial training and adaptive stepsizes. Our experiments over popular few-shot benchmarks show that our method yields increased accuracy and robustness to the choice of base stepsize. More generally, our results motivate the use of adversarial learning as a data augmentation scheme for improving few-shot generalization. In the future, it would be interesting to apply our method to related settings such as domain adaption and transfer learning.

### ACKNOWLEDGMENTS

We thank Hugo Larochelle and Reza Babanezhad for insightful discussions, Damien Scieur and Emmanuel Bengio for helpful feedback on the manuscript. We also thank the anonymous reviewers for their comments and suggestions.

This research was partially supported by the Canada CIFAR AI Chair Program, the NSERC Discovery Grant RGPIN-2017-06936 and a Google Focused Research award. Simon Lacoste-Julien is a CIFAR Associate Fellow in the Learning in Machines & Brains program.

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

## A APPENDIX

### A.1 NOTATIONS AND ACRONYMS

Notations and acronyms used repeatedly throughout the paper are summarized below.

| Notation | Meaning |
|---|---|
| USA | Uncertainty-based stepsize adaption |
| I/USA | Inverse USA |
| FGSM | Fast gradient sign method optimization (Goodfellow et al., 2015) |
| UFGSM | Uncertainty-based FGSM |
| I/UFGSM | Inverse UFGSM |
| EnAug | Ensemble augmentation |
| $T$ | Number of test-time gradient steps |
| $M$ | Size of deep ensemble |
| $\theta_0$ | Pretrained MAML checkpoint to repurpose |
| $\theta_t^m$ | Ensemble element $m$ at time $t$ |
| $\theta_T$ | Model parameters after meta-testing |
| $\alpha$ | Base stepsize |
| $\alpha^{\mathrm{adap}}$ | Uncertainty-based stepsize (parameter-wise) |
| $D^{\mathrm{Spt}}$ | Support set (test-time) |
| $D^{\mathrm{Aug}}$ | Adversarial examples based on $D^{\mathrm{Spt}}$ |

### A.2 DESIGN CHOICES FOR USA

Here, we introduce the design choices of the Algorithm 2.

**Line 3**: We calculate the maximum $\mathrm{Max}(u)$ and minimum $\mathrm{Min}(u)$ values among the estimated standard deviations $u$ for each parameter. As proposal 1, in order to assign to low stepsize to high uncertainty, the standard deviation of each parameter is subtracted from the maximum value $\mathrm{Max}(u) - u$ for inverse-relationship. At this time, the largest value among parameter values becomes 0. To prevent this, we add the smallest $\mathrm{Max}(u) - u + \mathrm{Min}(u)$. Note that all values of $u$ are non-negative. We also tried alternatives ways like non-inverse-relationship and relative standard deviation, but it did not seem to work better. To keep the same scale of the uncertainty before and after the transformation, we chose the inverse-relationship using $\mathrm{Max}(u) - u + \mathrm{Min}(u)$.

**Line 4**: In this study, we use layer-wise adapted stepsize. In order to assign the same stepsize for each layer, we calculate the average of the parameters of each layer and the values of the parameters of the corresponding layer are replaced with the average value. It repeats all layers and performs the corresponding operation.

**Line 5**: To more easily compare the effectiveness of USA with the baseline, we set the average stepsize equals to the base stepsize, i.e. $1/|\alpha^{\mathrm{adap}}| \sum \alpha^{\mathrm{adap}} = \alpha$. If USA is applied without this rescaling, than the average stepsize can change, which makes it difficult to distinguish the effect of using an overall different stepsize (for constant SGD e.g.) vs. our adaptation of stepsizes.

### A.3 EXPERIMENTAL SETUP AND PRETRAINED MODEL SELECTION

Our baseline model was trained using the same hyperparameters with *mini*ImageNet training of MAML except the inner loop stepsize. The inner loop stepsize was set to 0.1 for reproducing the 5-way 1-shot accuracy reported in the original MAML paper. We trained the model for 150,000 iterations. A pretrained model was selected with the validation accuracy among *mini*ImageNet training checkpoints. We used the checkpoint with the highest validation accuracy as the pretrained model $\theta_0$. The highest test accuracy we achieved was 49.24% during meta-training. For the checkpoint that we chose with the highest validation performance, the test accuracy was 47.58%. Note that the cross-domain performance highly depended on which checkpoint we used. See Fig.4 in Appendix A.4.

The size of ensemble is $M = 5$ which is the same as the deep ensembles (Lakshminarayanan et al., 2017). The parameter for the FGSM is $\epsilon = 0.05$. The scale value a of Gaussian random perturbation for ensemble model training is $\sigma = 0.05$. The gradient step is $T = 10$ which is same with *mini*ImageNet test of MAML.

We do not split those datasets except *mini*ImageNet, because we do not use the datasets in the meta-training. We used the same splits as Ravi & Larochelle (2017) for *mini*ImageNet. Tseng et al. (2020) used a randomly split manner for the cross-domain test and Triantafillou et al. (2020) used all traffic signs dataset for the test.

### A.4   CROSS-DOMAIN ACCURACY WHILE META-TRAINING MAML ON *MINI*IMAGENET

The performance of the cross-domain significantly differed depending on the selected checkpoint. Figure 4 shows the performance for every 200 iterations. We selected the checkpoint based on miniImageNet validation performance, which is the highest at the 57,200 iteration.

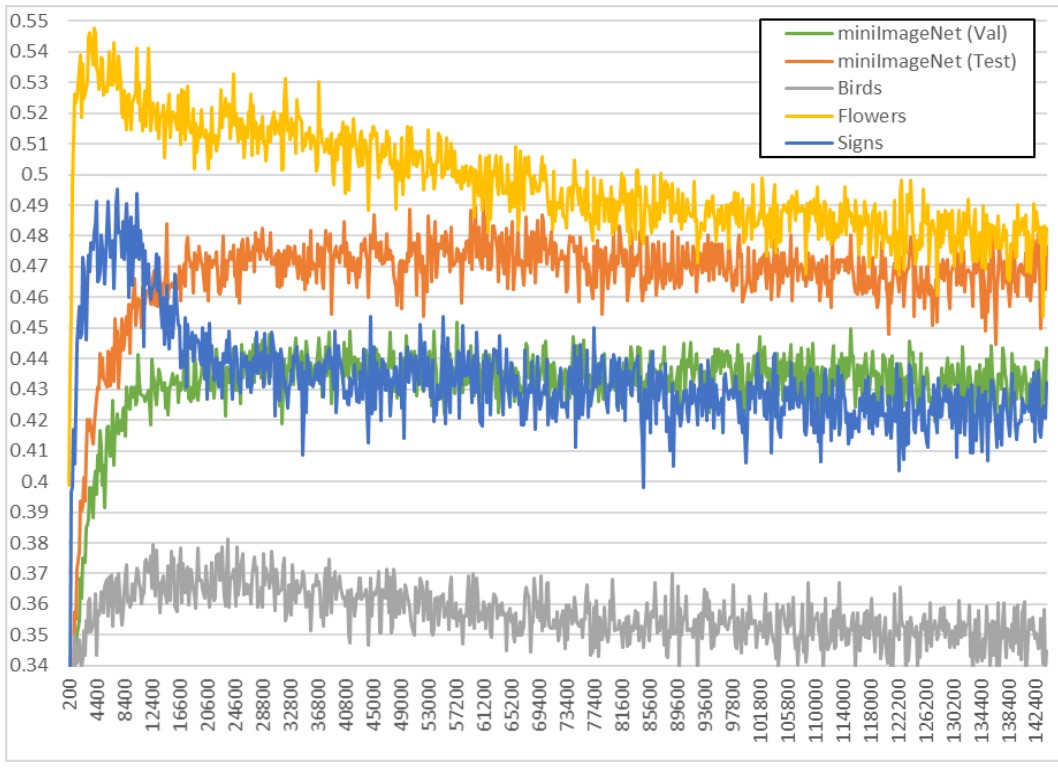

Figure 4: Cross-domain performance while meta-training on *mini*ImageNet. X-axis is training iteration on meta-training on *mini*ImageNet and y-axis is classification accuracy.

### A.5   ADDITIONAL RESULTS FOR AT-ONLY SETTING.

To evaluate the effect of AT, we only applied AT in MAML ($\lambda_\alpha = \lambda_{\text{AT}} = 1, \lambda_{\text{Aug}} = 0$). **Table 5**, **Figure 5** and **6** show AT results on 5-way 1-shot with SGD and Adam. Adam is worse than SGD in the results. However, we found that AT and Adam are good combination for a meta-test training.

### A.6   ADDITIONAL RESULTS FOR 5-WAY 5-SHOT AND 10-WAY 1-SHOT CLASSIFICATION.

We present some additional results for 5-way 5-shot and 10-way 1-shot classification in this section. **Table 6** shows the 5-way 5-shot and 10-way 1-shot classification results. Especially in the case of the Top-40% Avg our method significantly increased the performance about 5.84% and 1.83% in 5-way 5-shot and 10-way 1-shot respectively. It means that it can increase the probability to select of

Table 5: 5-way 1-shot classification results. In order to verify the effectiveness of AT, we applied only AT among the proposed method and the all proposed method.

| Metric | Dataset | Baseline SGD | Ours (SGD) AT Only | All | Baseline Adam | Ours (Adam) AT Only | All |
|---|---|---|---|---|---|---|---|
| All | Mini | $\mathbf{37.21}_{\pm \mathbf{0.22}}$ | $34.60_{\pm 0.18}$ | $36.24_{\pm 0.22}$ | $31.87_{\pm 0.20}$ | $33.40_{\pm 0.33}$ | $\mathbf{33.57}_{\pm \mathbf{0.29}}$ |
| | Birds | $\mathbf{30.07}_{\pm \mathbf{0.52}}$ | $28.81_{\pm 0.45}$ | $29.83_{\pm 0.46}$ | $28.02_{\pm 0.39}$ | $29.37_{\pm 0.51}$ | $\mathbf{29.51}_{\pm \mathbf{0.46}}$ |
| | Flowers | $40.54_{\pm 0.48}$ | $38.80_{\pm 0.44}$ | $\mathbf{40.86}_{\pm \mathbf{0.45}}$ | $40.33_{\pm 0.57}$ | $44.53_{\pm 0.68}$ | $\mathbf{44.58}_{\pm \mathbf{0.70}}$ |
| | Signs | $34.18_{\pm 0.76}$ | $33.42_{\pm 0.59}$ | $\mathbf{34.76}_{\pm \mathbf{0.61}}$ | $33.98_{\pm 0.43}$ | $34.96_{\pm 0.47}$ | $\mathbf{35.10}_{\pm \mathbf{0.43}}$ |
| | Avg. | $\mathbf{35.50}_{\pm \mathbf{0.49}}$ | $33.91_{\pm 0.41}$ | $35.42_{\pm 0.44}$ | $33.55_{\pm 0.22}$ | $35.57_{\pm 0.29}$ | $\mathbf{35.69}_{\pm \mathbf{0.29}}$ |
| Top-1 | Mini | $\mathbf{46.71}_{\pm \mathbf{0.81}}$ | $46.58_{\pm 0.65}$ | $46.62_{\pm 0.58}$ | $43.08_{\pm 0.10}$ | $\mathbf{44.80}_{\pm \mathbf{0.40}}$ | $44.72_{\pm 0.35}$ |
| | Birds | $36.03_{\pm 0.48}$ | $36.40_{\pm 0.48}$ | $\mathbf{36.43}_{\pm \mathbf{0.58}}$ | $34.00_{\pm 0.63}$ | $35.28_{\pm 0.66}$ | $\mathbf{35.45}_{\pm \mathbf{0.67}}$ |
| | Flowers | $51.97_{\pm 0.79}$ | $\mathbf{54.55}_{\pm \mathbf{0.84}}$ | $54.36_{\pm 1.04}$ | $51.28_{\pm 0.47}$ | $\mathbf{56.10}_{\pm \mathbf{1.06}}$ | $56.04_{\pm 1.14}$ |
| | Signs | $43.44_{\pm 0.99}$ | $43.98_{\pm 0.98}$ | $\mathbf{44.10}_{\pm \mathbf{1.03}}$ | $42.53_{\pm 0.79}$ | $\mathbf{43.13}_{\pm \mathbf{0.83}}$ | $43.01_{\pm 0.85}$ |
| | Avg. | $44.54_{\pm 0.77}$ | $\mathbf{45.38}_{\pm \mathbf{0.74}}$ | $45.37_{\pm 0.81}$ | $42.72_{\pm 0.32}$ | $\mathbf{44.83}_{\pm \mathbf{0.51}}$ | $44.80_{\pm 0.54}$ |
| Top-40% | Mini | $45.07_{\pm 0.52}$ | $45.53_{\pm 0.50}$ | $\mathbf{45.79}_{\pm \mathbf{0.44}}$ | $40.18_{\pm 0.14}$ | $42.59_{\pm 0.43}$ | $\mathbf{42.92}_{\pm \mathbf{0.35}}$ |
| | Birds | $34.88_{\pm 0.55}$ | $35.57_{\pm 0.60}$ | $\mathbf{35.77}_{\pm \mathbf{0.55}}$ | $32.38_{\pm 0.61}$ | $34.26_{\pm 0.73}$ | $\mathbf{34.54}_{\pm \mathbf{0.66}}$ |
| | Flowers | $49.98_{\pm 0.69}$ | $52.92_{\pm 0.68}$ | $\mathbf{53.16}_{\pm \mathbf{0.73}}$ | $48.18_{\pm 0.44}$ | $53.88_{\pm 0.72}$ | $\mathbf{54.08}_{\pm \mathbf{0.76}}$ |
| | Signs | $41.81_{\pm 1.02}$ | $43.19_{\pm 0.96}$ | $\mathbf{43.39}_{\pm \mathbf{1.01}}$ | $40.90_{\pm 0.70}$ | $42.06_{\pm 0.74}$ | $\mathbf{42.26}_{\pm \mathbf{0.77}}$ |
| | Avg. | $42.94_{\pm 0.70}$ | $44.30_{\pm 0.68}$ | $\mathbf{44.52}_{\pm \mathbf{0.68}}$ | $40.41_{\pm 0.29}$ | $43.20_{\pm 0.42}$ | $\mathbf{43.45}_{\pm \mathbf{0.40}}$ |

(a) Mini    (b) Birds    (c) Flowers    (d) Signs

Figure 5: 5-way 1-shot classification results with AT on SGD.

optimal stepsize for a new task. As shown in Figure 7, there are more flatten curve near the highest accuracy in both tasks.

## A.7 VALIDATING PROPOSALS 1 AND 2 WITH "INVERSE" COUNTERPARTS.

To validate proposal 1, we evaluate the inverse strategy of USA, denoted I/USA, which assigns higher stepsizes to layers with higher uncertainty. Specifically, we flip the stepsizes by replacing **line 3** of Algorithm 2 with $c \leftarrow u$. To measure the effect of USA, we only applied USA in MAML ($\lambda_\alpha = \lambda_{AT} = \lambda_{Aug} = 0$). The results for 5-way 1-shot classification are in **Table 7**. USA outperforms the baselines over broad ranges of stepsizes compared to Adam and SGD. I/USA significantly decreased the performance of all metrics. Therefore, USA reflected useful knowledge well into the stepsize for a new task. See accuracy by stepsizes in Fig 8. Figure 8 shows the results for verifying of Proposal 1. USA shows more flatten curve in Top-1 accuracy ranges than SGD and I/USA. I/USA degraded the performance through all ranges of stepsizes.

For verifying the proposal 2, we evaluated three methods UFGSM , I/UFGSM and FGSM on USA. I/UFGSM is inversely implemented method of UFGSM. FGSM uses examples generated by FGSM instead of UFGSM. To measure the effect of UFGSM, we only applied UFGSM in MAML with USA ($\lambda_\alpha = \lambda_{AT} = 0, \lambda_{Aug} = 1$). Table 7 shows 5-way 1-shot classification results. UFGSM outperforms all the other methods for every metric. In spite of I/UFGSM had stronger adversarial perturbation than UFGSM, the performance was worse than UFGSM. Note that I/UFGSM and FGSM showed almost similar performances. The reason is that most of the input pixels have an uncertainty close to 0; therefore, when scaling after inverse, most pixels have a value of 1. Therefore, the generated examples are almost similar to the adversarial example in which FGSM is applied. See accuracy by stepsizes in Fig 9. Figure 9 shows the result for verifying of proposal 2. UFGSM

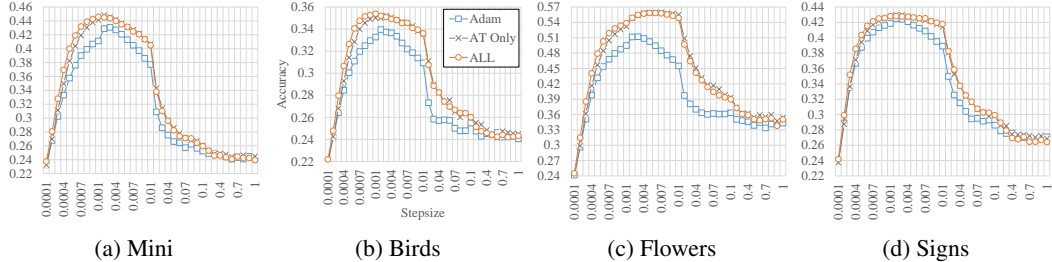

Figure 6: 5-way 1-shot classification results with AT on Adam.

Table 6: 5-way 5-shot and 10-way 1-shot classification results using SGD. Our method outperformed both tasks. Especially 5-way 5-shot classification performance significantly improved than baseline.

| Metric | Dataset | 5-way 5-shot Baseline SGD | Ours (SGD) SGD+AT | SGD+All | 10-way 1-shot Baseline SGD | Ours (SGD) SGD+AT | SGD+All |
|--------|---------|---------|---------|---------|---------|---------|---------|
| All | Mini | $38.14_{\pm 0.39}$ | $41.06_{\pm 0.35}$ | $\mathbf{43.48_{\pm 0.38}}$ | $21.31_{\pm 0.37}$ | $20.97_{\pm 0.37}$ | $\mathbf{22.07_{\pm 0.40}}$ |
| | Birds | $33.29_{\pm 0.29}$ | $36.04_{\pm 0.28}$ | $\mathbf{38.07_{\pm 0.33}}$ | $17.54_{\pm 0.06}$ | $17.08_{\pm 0.07}$ | $\mathbf{17.81_{\pm 0.12}}$ |
| | Flowers | $44.55_{\pm 0.43}$ | $48.94_{\pm 0.48}$ | $\mathbf{53.10_{\pm 0.55}}$ | $26.79_{\pm 0.41}$ | $26.50_{\pm 0.42}$ | $\mathbf{28.07_{\pm 0.41}}$ |
| | Signs | $45.39_{\pm 0.45}$ | $47.98_{\pm 0.55}$ | $\mathbf{51.02_{\pm 0.50}}$ | $23.77_{\pm 0.13}$ | $23.99_{\pm 0.11}$ | $\mathbf{24.88_{\pm 0.13}}$ |
| | Avg. | $40.34_{\pm 0.39}$ | $43.51_{\pm 0.41}$ | $\mathbf{46.42_{\pm 0.44}}$ | $22.35_{\pm 0.24}$ | $22.14_{\pm 0.24}$ | $\mathbf{23.21_{\pm 0.26}}$ |
| Top-1 | Mini | $\mathbf{62.46_{\pm 0.68}}$ | $62.07_{\pm 0.61}$ | $62.16_{\pm 0.66}$ | $31.10_{\pm 0.66}$ | $31.18_{\pm 0.81}$ | $\mathbf{31.42_{\pm 0.82}}$ |
| | Birds | $50.71_{\pm 0.54}$ | $\mathbf{51.93_{\pm 0.62}}$ | $51.82_{\pm 0.64}$ | $23.16_{\pm 0.21}$ | $23.42_{\pm 0.37}$ | $\mathbf{23.61_{\pm 0.39}}$ |
| | Flowers | $70.63_{\pm 0.76}$ | $74.14_{\pm 0.72}$ | $\mathbf{74.52_{\pm 0.69}}$ | $38.61_{\pm 0.49}$ | $\mathbf{40.84_{\pm 0.58}}$ | $40.68_{\pm 0.58}$ |
| | Signs | $72.52_{\pm 0.84}$ | $73.66_{\pm 0.98}$ | $\mathbf{74.13_{\pm 0.90}}$ | $35.10_{\pm 0.21}$ | $34.76_{\pm 0.17}$ | $\mathbf{35.11_{\pm 0.26}}$ |
| | Avg. | $64.08_{\pm 0.71}$ | $65.45_{\pm 0.73}$ | $\mathbf{65.66_{\pm 0.72}}$ | $31.99_{\pm 0.39}$ | $32.55_{\pm 0.48}$ | $\mathbf{32.70_{\pm 0.51}}$ |
| Top-40% | Mini | $55.38_{\pm 0.46}$ | $59.97_{\pm 0.69}$ | $\mathbf{60.70_{\pm 0.91}}$ | $28.65_{\pm 0.64}$ | $30.22_{\pm 0.74}$ | $\mathbf{30.56_{\pm 0.77}}$ |
| | Birds | $45.50_{\pm 0.49}$ | $50.48_{\pm 0.53}$ | $\mathbf{51.03_{\pm 0.63}}$ | $21.78_{\pm 0.07}$ | $22.61_{\pm 0.22}$ | $\mathbf{22.87_{\pm 0.30}}$ |
| | Flowers | $65.16_{\pm 0.73}$ | $72.61_{\pm 0.78}$ | $\mathbf{73.55_{\pm 0.86}}$ | $36.14_{\pm 0.53}$ | $39.40_{\pm 0.66}$ | $\mathbf{39.75_{\pm 0.61}}$ |
| | Traffic | $67.34_{\pm 0.43}$ | $70.56_{\pm 0.87}$ | $\mathbf{71.45_{\pm 0.90}}$ | $33.56_{\pm 0.11}$ | $34.21_{\pm 0.13}$ | $\mathbf{34.27_{\pm 0.17}}$ |
| | Avg. | $58.34_{\pm 0.53}$ | $63.40_{\pm 0.72}$ | $\mathbf{64.18_{\pm 0.82}}$ | $30.03_{\pm 0.34}$ | $31.61_{\pm 0.44}$ | $\mathbf{31.86_{\pm 0.46}}$ |

shows flatter curve near the best performing accuracy than SGD, I/UFGSM and FGSM. FGSM and I/UFGSM show very similar trends. We explained the reason why the two methods gave similar results. As can be seen in Fig 10, almost all of the input gradient uncertainty is near zero. When we inverse the value, almost all the value is 1 (See Fig 11). UFGSM improves performance despite less adversarial perturbation than I/UFGSM and FGSM. Through this, even a small change can help model learning if correct (useful) information is reflected.

## A.8    5-WAY 5-SHOT CLASSIFICATION RESULTS ON OVERFITTED CHECKPOINT

As can be seen in Appendix A.4, overfitted models on *mini*ImageNet degraded performance over all datasets. We investigated the 5-way 5-shot classification results on the checkpoint after meta-training with MAML for 150K iterations. **Table 8** and **9** show the results with SGD and Adam. The number in the parentheses is the difference from the baseline for each checkpoint (e.g. SGD or Adam). The results using the (Last) checkpoint are worse than (Validation) checkpoint due to the overfitting. However the performance boost from using our method is more substantial on the over-fitted (Last) checkpoint than the best (Validation) checkpoint, both in terms of absolute performance (Top-1) and robustness to choice of base stepsize (other metrics).

## A.9    FREEZING MOST UNCERTAIN LAYERS.

Since some of the BatchNorm parameters have the highest uncertainty (see Figure 1a), we have experimented freezing the BatchNorm parameters during meta-testing (i.e. setting their stepsize to zero). It turns out that SGD+All (w/ freezing BN) outperforms SGD+All (w/o freezing BN) on the All metric (**Table 10**), with most improvement in the higher stepsize range (>0.1) (**Figure 12**). We

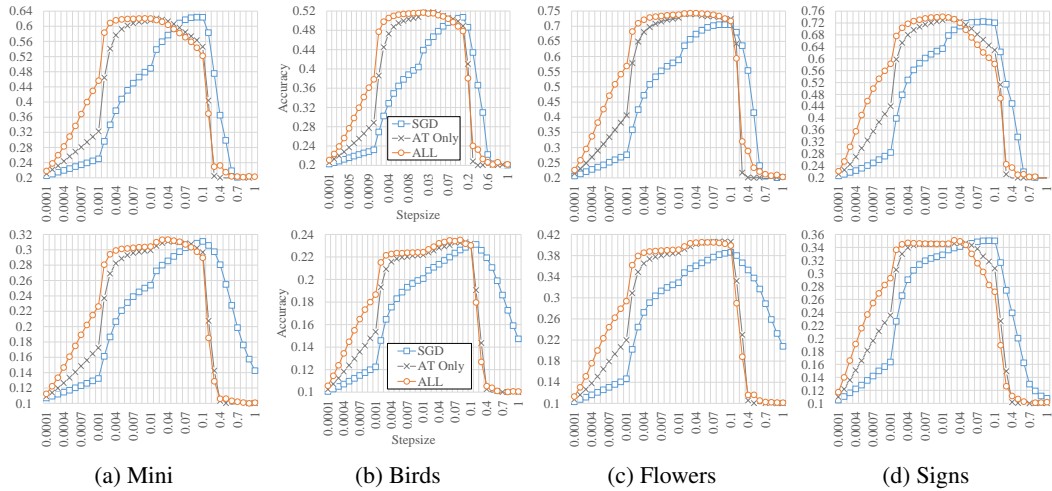

Figure 7: 5-way 5-shot (Top) and 10-way 1-shot (Bottom) classification with MAML(SGD) and our proposed method. In terms of robustness, our method outperforms MAML(SGD) more effectively. In addition, we show that our method outperforms Top-1 accuracy for Birds, Flowers and Signs. Also our method shows flatter curve near highest accuracy sections. This increases the probability of optimal stepsize selection.

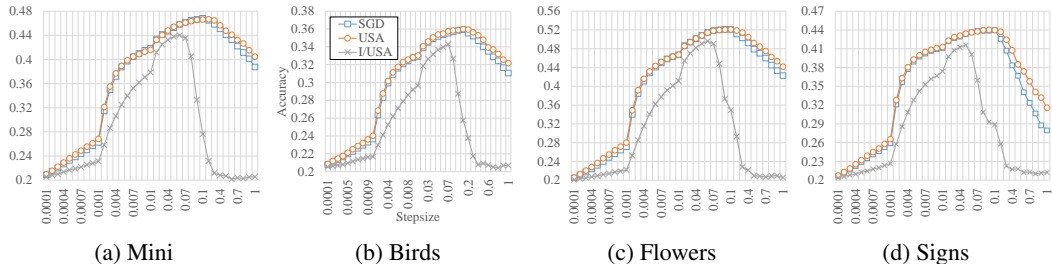

Figure 8: 5-way 1-shot classification for verifying proposal 1. We compared to USA and I/USA.

notice a similar trend for 5-way 5-shot SGD, where SGD (w/ freezing BN) also outperforms SGD (w/o freezing BN) on the All metric, with most improvement in the higher stepsize range.

Table 7: 5-way 1-shot classification results for verifying proposal 1 and 2. I/USA and I/UFGSM degraded performance than USA and UFGSM respectively. Our approach which is applied USA and UFGSM outperformed SGD on almost of the metrics.

| Metric | Dataset | Baseline | | SGD | | SGD+USA | | |
|---|---|---|---|---|---|---|---|---|
| | | SGD | Adam | +USA | +I/USA | +UFGSM | +I/UFGSM | +FGSM |
| All | Mini | 37.21 | 31.85 | 37.53 | 28.73 | **38.71** | 37.25 | 36.97 |
| | Birds | 30.07 | 28.17 | 30.39 | 24.98 | **31.07** | 30.27 | 30.05 |
| | Flowers | 40.54 | 40.49 | 41.09 | 30.52 | **42.33** | 41.44 | 41.17 |
| | Signs | 34.21 | 34.28 | 35.02 | 27.90 | **36.10** | 34.98 | 34.78 |
| | Avg. | 35.50 | 33.68 | 36.01 | 28.03 | **37.05** | 35.98 | 35.74 |
| Top-1 | Mini | 46.71 | 43.07 | **46.72** | 44.13 | 46.66 | 44.31 | 43.93 |
| | Birds | 36.03 | 34.30 | **36.08** | 34.20 | 36.02 | 34.77 | 34.52 |
| | Flowers | 51.97 | 51.38 | 52.01 | 49.34 | **52.30** | 50.90 | 50.63 |
| | Signs | 43.44 | 43.00 | 43.42 | 41.03 | **43.96** | 42.82 | 42.62 |
| | Avg. | 44.54 | 42.93 | 44.56 | 42.17 | **44.73** | 43.20 | 42.93 |
| Top-40% | Mini | 45.07 | 40.16 | 45.23 | 38.33 | **45.76** | 43.60 | 43.18 |
| | Birds | 34.88 | 32.64 | 35.17 | 30.28 | **35.30** | 34.20 | 33.88 |
| | Flowers | 49.98 | 48.28 | 50.39 | 42.19 | **51.07** | 49.79 | 49.47 |
| | Signs | 41.81 | 41.32 | 42.08 | 35.78 | **43.18** | 42.17 | 41.92 |
| | Avg. | 42.94 | 40.60 | 43.22 | 36.64 | **43.83** | 42.44 | 42.11 |

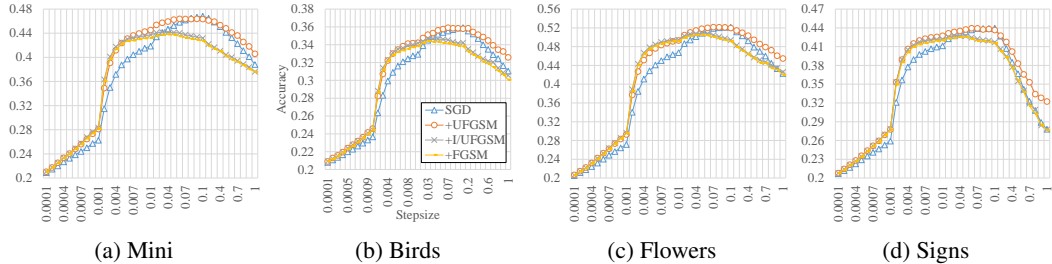

(a) Mini      (b) Birds      (c) Flowers      (d) Signs

Figure 9: 5-way 1-shot classification results to verify proposal 2. We compared to UFGSM, I/UFGSM and FGSM.

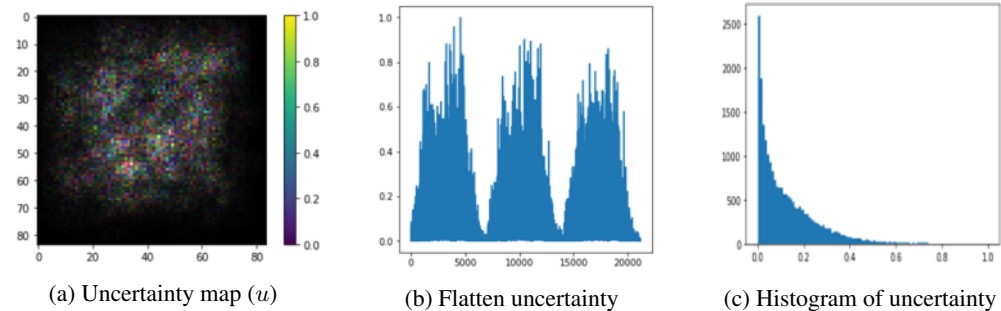

(a) Uncertainty map ($u$)      (b) Flatten uncertainty      (c) Histogram of uncertainty

Figure 10: (a) shows the re-scaled input gradient uncertainty to generate the UFGSM example. (b) is a flatten plot of the re-scaled input gradient uncertainty. (c) shows a histogram of the uncertainty.

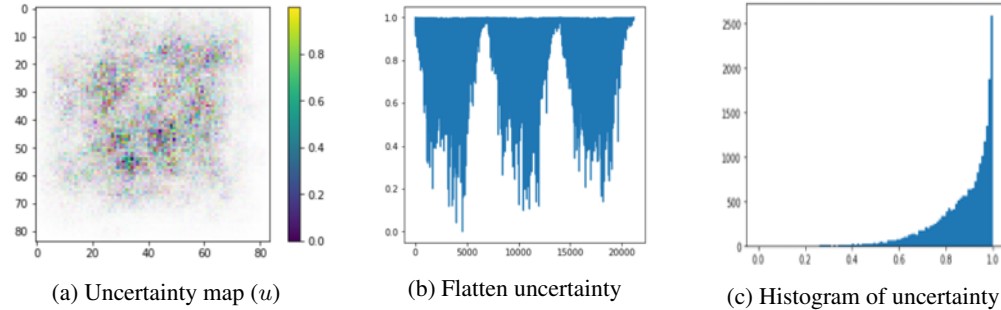

(a) Uncertainty map ($u$)  (b) Flatten uncertainty  (c) Histogram of uncertainty

Figure 11: (a) shows the re-scaled inverse input gradient uncertainty to generate the UFGSM example. (b) is a flatten plot of the re-scaled inverse input gradient uncertainty. (c) shows a histogram of the uncertainty.

Table 8: 5-way 5-shot classification results between selected checkpoint with SGD

| Metric | Dataset | Checkpoint (Validation) | | | Checkpoint (Last) | | |
|---|---|---|---|---|---|---|---|
| | | Baseline | Ours (SGD) | | Baseline | Ours (SGD) | |
| | | SGD | AT Only | All | SGD | AT Only | All |
| All | Mini | $38.14_{\pm0.39}$ | $41.06_{\pm0.35}$ | $\mathbf{43.48}_{\pm\mathbf{0.38}}$ | $34.98_{\pm0.54}$ | $38.66_{\pm0.52}$ | $41.89_{\pm0.61}$ |
| | | (-) | (2.92) | (5.34) | (-) | (3.68) | **(6.91)** |
| | Birds | $33.29_{\pm0.29}$ | $36.04_{\pm0.28}$ | $\mathbf{38.07}_{\pm\mathbf{0.33}}$ | $31.33_{\pm0.51}$ | $34.34_{\pm0.49}$ | $36.90_{\pm0.5}$ |
| | | (-) | (2.75) | (4.78) | (-) | (3.01) | **(5.57)** |
| | Flowers | $44.55_{\pm0.43}$ | $48.94_{\pm0.48}$ | $\mathbf{53.10}_{\pm\mathbf{0.55}}$ | $40.96_{\pm0.67}$ | $45.6_{\pm0.6}$ | $50.58_{\pm0.67}$ |
| | | (-) | (4.39) | (8.55) | (-) | (4.64) | **(9.62)** |
| | Signs | $45.39_{\pm0.45}$ | $47.98_{\pm0.55}$ | $\mathbf{51.02}_{\pm\mathbf{0.50}}$ | $42.05_{\pm0.35}$ | $46.01_{\pm0.36}$ | $49.88_{\pm0.35}$ |
| | | (-) | (2.59) | (5.63) | (-) | (3.96) | **(7.83)** |
| | Avg. | $40.34_{\pm0.39}$ | $43.51_{\pm0.41}$ | $\mathbf{46.42}_{\pm\mathbf{0.44}}$ | $37.33_{\pm0.52}$ | $41.15_{\pm0.49}$ | $44.81_{\pm0.53}$ |
| | | (-) | (3.17) | (6.08) | (-) | (3.82) | **(7.48)** |
| Top-1 | Mini | $\mathbf{62.46}_{\pm\mathbf{0.68}}$ | $62.07_{\pm0.61}$ | $62.16_{\pm0.66}$ | $58.49_{\pm0.72}$ | $58.44_{\pm0.93}$ | $58.72_{\pm1.08}$ |
| | | (-) | (-0.39) | (-0.3) | (-) | (-0.05) | **(0.23)** |
| | Birds | $50.71_{\pm0.54}$ | $\mathbf{51.93}_{\pm\mathbf{0.62}}$ | $51.82_{\pm0.64}$ | $46.77_{\pm0.53}$ | $48.33_{\pm0.72}$ | $48.58_{\pm0.7}$ |
| | | (-) | (1.22) | (1.11) | (-) | (1.56) | **(1.81)** |
| | Flowers | $70.63_{\pm0.76}$ | $74.14_{\pm0.72}$ | $\mathbf{74.52}_{\pm\mathbf{0.69}}$ | $66.31_{\pm1.02}$ | $68.99_{\pm0.81}$ | $69.73_{\pm0.71}$ |
| | | (-) | (3.51) | **(3.89)** | (-) | (2.68) | (3.42) |
| | Signs | $72.52_{\pm0.84}$ | $73.66_{\pm0.98}$ | $\mathbf{74.13}_{\pm\mathbf{0.90}}$ | $67.75_{\pm0.57}$ | $70.71_{\pm0.48}$ | $71.08_{\pm0.53}$ |
| | | (-) | (1.14) | (1.61) | (-) | (2.96) | **(3.33)** |
| | Avg. | $64.08_{\pm0.71}$ | $65.45_{\pm0.73}$ | $\mathbf{65.66}_{\pm\mathbf{0.72}}$ | $59.83_{\pm0.71}$ | $61.62_{\pm0.73}$ | $62.03_{\pm0.75}$ |
| | | (-) | (1.37) | (1.58) | (-) | (1.79) | **(2.2)** |
| Top-40% | Mini | $55.38_{\pm0.46}$ | $59.97_{\pm0.69}$ | $\mathbf{60.70}_{\pm\mathbf{0.91}}$ | $50.56_{\pm0.72}$ | $56.44_{\pm0.87}$ | $57.22_{\pm1}$ |
| | | (-) | (4.59) | (5.32) | (-) | (5.88) | **(6.66)** |
| | Birds | $45.50_{\pm0.49}$ | $50.48_{\pm0.53}$ | $\mathbf{51.03}_{\pm\mathbf{0.63}}$ | $42.13_{\pm0.55}$ | $47.22_{\pm0.73}$ | $47.98_{\pm0.73}$ |
| | | (-) | (4.98) | (5.53) | (-) | (5.09) | **(5.85)** |
| | Flowers | $65.16_{\pm0.73}$ | $72.61_{\pm0.78}$ | $\mathbf{73.55}_{\pm\mathbf{0.86}}$ | $59.8_{\pm1.02}$ | $67.37_{\pm0.83}$ | $68.93_{\pm0.79}$ |
| | | (-) | (7.45) | (8.39) | (-) | (7.57) | **(9.13)** |
| | Signs | $67.34_{\pm0.43}$ | $70.56_{\pm0.87}$ | $\mathbf{71.45}_{\pm\mathbf{0.90}}$ | $61.95_{\pm0.66}$ | $68.03_{\pm0.64}$ | $69.41_{\pm0.5}$ |
| | | (-) | (3.22) | (4.11) | (-) | (6.08) | **(7.46)** |
| | Avg. | $58.34_{\pm0.53}$ | $63.40_{\pm0.72}$ | $\mathbf{64.18}_{\pm\mathbf{0.82}}$ | $53.61_{\pm0.74}$ | $59.77_{\pm0.77}$ | $60.89_{\pm0.76}$ |
| | | (-) | (5.06) | (5.84) | (-) | (6.16) | **(7.28)** |

Table 9: 5-way 5-shot classification results of selected checkpoint with Adam

| Metric | Dataset | Checkpoint (Validation) | | | Checkpoint (Last) | | |
| | | Baseline Adam | Ours (Adam) AT Only | All | Baseline Adam | Ours (Adam) AT Only | All |
|---|---|---|---|---|---|---|---|
| All | Mini | $40.08_{\pm1.33}$ (-) | $41.38_{\pm1.07}$ (1.3) | $\mathbf{41.64_{\pm0.97}}$ (1.56) | $37.25_{\pm0.64}$ (-) | $39.47_{\pm0.56}$ (2.21) | $40.36_{\pm0.62}$ (**3.1**) |
| | Birds | $36.68_{\pm1.05}$ (-) | $\mathbf{38.79_{\pm1}}$ (2.11) | $38.78_{\pm0.91}$ (2.10) | $35.39_{\pm0.67}$ (-) | $37.86_{\pm0.72}$ (2.47) | $38.15_{\pm0.69}$ (**2.76**) |
| | Flowers | $58.00_{\pm1.55}$ (-) | $\mathbf{60.92_{\pm1.24}}$ (2.92) | $60.36_{\pm1.14}$ (2.36) | $54.36_{\pm0.79}$ (-) | $59.09_{\pm0.63}$ (4.73) | $59.32_{\pm0.69}$ (**4.96**) |
| | Signs | $\mathbf{52.65_{\pm0.57}}$ (-) | $52.37_{\pm0.6}$ (-0.28) | $52.53_{\pm0.64}$ (-0.12) | $50.48_{\pm0.6}$ (-) | $51.13_{\pm0.45}$ (0.65) | $51.95_{\pm0.49}$ (**1.47**) |
| | Avg. | $46.85_{\pm1.12}$ (-) | $\mathbf{48.37_{\pm0.98}}$ (1.52) | $48.33_{\pm0.91}$ (1.48) | $44.37_{\pm0.67}$ (-) | $46.89_{\pm0.59}$ (2.52) | $47.45_{\pm0.62}$ (**3.08**) |
| Top-1 | Mini | $58.89_{\pm0.73}$ (-) | $\mathbf{60.35_{\pm0.83}}$ (1.46) | $60.08_{\pm0.83}$ (1.19) | $55.25_{\pm1.02}$ (-) | $57.79_{\pm0.81}$ (2.54) | $57.86_{\pm0.92}$ (**2.61**) |
| | Birds | $47.92_{\pm0.75}$ (-) | $\mathbf{50.88_{\pm0.66}}$ (2.96) | $50.84_{\pm0.75}$ (2.92) | $46.02_{\pm0.85}$ (-) | $49.89_{\pm1}$ (**3.87**) | $49.78_{\pm0.97}$ (3.76) |
| | Flowers | $73.21_{\pm1.19}$ (-) | $\mathbf{77.51_{\pm0.94}}$ (4.3) | $77.45_{\pm0.89}$ (4.24) | $69.09_{\pm0.77}$ (-) | $76.61_{\pm0.51}$ (**7.52**) | $76.51_{\pm0.71}$ (7.42) |
| | Signs | $70.98_{\pm1.07}$ (-) | $71.30_{\pm1.52}$ (0.32) | $\mathbf{71.72_{\pm1.52}}$ (0.74) | $67.51_{\pm0.72}$ (-) | $69.75_{\pm0.83}$ (2.24) | $70.30_{\pm0.79}$ (**2.79**) |
| | Avg. | $62.75_{\pm0.93}$ (-) | $65.01_{\pm0.99}$ (2.26) | $\mathbf{65.02_{\pm1}}$ (2.27) | $59.47_{\pm0.84}$ (-) | $63.51_{\pm0.79}$ (4.04) | $63.61_{\pm0.85}$ (**4.14**) |
| Top-40% | Mini | $52.17_{\pm1.89}$ (-) | $54.96_{\pm1.18}$ (2.79) | $\mathbf{55.75_{\pm1.07}}$ (3.58) | $46.95_{\pm0.82}$ (-) | $51.81_{\pm0.71}$ (4.86) | $53.29_{\pm0.76}$ (**6.34**) |
| | Birds | $43.88_{\pm1.06}$ (-) | $47.93_{\pm0.91}$ (4.05) | $\mathbf{48.26_{\pm0.87}}$ (4.38) | $41.41_{\pm0.8}$ (-) | $46.71_{\pm0.96}$ (5.3) | $47.11_{\pm0.88}$ (**5.7**) |
| | Flowers | $68.43_{\pm1.76}$ (-) | $73.86_{\pm1.02}$ (5.43) | $\mathbf{74.03_{\pm1.02}}$ (5.60) | $63.44_{\pm0.77}$ (-) | $72.41_{\pm0.79}$ (8.97) | $72.56_{\pm0.64}$ (**9.12**) |
| | Signs | $66.85_{\pm0.78}$ (-) | $67.36_{\pm1.3}$ (0.51) | $\mathbf{67.89_{\pm1.39}}$ (1.04) | $62.95_{\pm0.81}$ (-) | $65.43_{\pm0.68}$ (2.48) | $66.37_{\pm0.71}$ (**3.42**) |
| | Avg. | $57.83_{\pm1.37}$ (-) | $61.03_{\pm1.1}$ (3.2) | $\mathbf{61.48_{\pm1.09}}$ (3.65) | $53.69_{\pm0.8}$ (-) | $59.09_{\pm0.78}$ (5.4) | $59.83_{\pm0.75}$ (**6.14**) |

(a) Mini  (b) Birds  (c) Flowers  (d) Signs

Figure 12: 5-way 1-shot (Top) and 5-way 5-shot (Bottom) classification results of freezing BN layers. We tested freezing BN on SGD+All. SGD+All (w/ BN freezing) outperformed SGD+All (w/o BN freezing) in the higher stepsize range (>0.1).

Table 10: 5-way 1-shot and 5-way 5-shot classification results of freezing BN layers. We tested freezing BN on our SGD+All. SGD+All (w/ freezing BN) outperformed in All metric on 5-way 1-shot and 5-way 5-shot.

| Metric | Dataset | 5-way 1-shot (SGD+All) | | 5-way 5-shot (SGD+All) | |
|---|---|---|---|---|---|
| | | w/ freezing BN | w/o freezing BN | w/ freezing BN | w/o freezing BN |
| All | Mini | $\mathbf{37.13}_{\pm 0.24}$ | $36.24_{\pm 0.22}$ | $\mathbf{45.42}_{\pm 0.42}$ | $43.48_{\pm 0.38}$ |
| | Birds | $\mathbf{30.74}_{\pm 0.50}$ | $29.83_{\pm 0.46}$ | $\mathbf{40.29}_{\pm 0.47}$ | $38.07_{\pm 0.33}$ |
| | Flowers | $\mathbf{43.88}_{\pm 0.56}$ | $40.86_{\pm 0.45}$ | $\mathbf{59.41}_{\pm 0.68}$ | $53.10_{\pm 0.55}$ |
| | Signs | $\mathbf{36.37}_{\pm 0.67}$ | $34.76_{\pm 0.61}$ | $\mathbf{54.65}_{\pm 0.56}$ | $51.02_{\pm 0.50}$ |
| | Avg. | $\mathbf{37.03}_{\pm 0.49}$ | $35.42_{\pm 0.44}$ | $\mathbf{49.94}_{\pm 0.54}$ | $46.42_{\pm 0.44}$ |
| Top-1 | Mini | $46.48_{\pm 0.54}$ | $46.62_{\pm 0.58}$ | $62.18_{\pm 0.66}$ | $62.16_{\pm 0.66}$ |
| | Birds | $36.41_{\pm 0.59}$ | $36.43_{\pm 0.58}$ | $51.90_{\pm 0.73}$ | $51.82_{\pm 0.64}$ |
| | Flowers | $54.16_{\pm 0.80}$ | $54.36_{\pm 1.04}$ | $74.59_{\pm 0.79}$ | $74.52_{\pm 0.69}$ |
| | Signs | $44.18_{\pm 1.09}$ | $44.10_{\pm 1.03}$ | $73.98_{\pm 0.88}$ | $74.13_{\pm 0.90}$ |
| | Avg. | $45.31_{\pm 0.76}$ | $45.37_{\pm 0.81}$ | $65.66_{\pm 0.76}$ | $65.66_{\pm 0.72}$ |
| Top-40% | Mini | $45.44_{\pm 0.48}$ | $45.79_{\pm 0.44}$ | $60.62_{\pm 0.66}$ | $60.70_{\pm 0.91}$ |
| | Birds | $35.67_{\pm 0.54}$ | $35.77_{\pm 0.55}$ | $51.04_{\pm 0.73}$ | $51.03_{\pm 0.63}$ |
| | Flowers | $52.97_{\pm 0.71}$ | $53.16_{\pm 0.73}$ | $73.56_{\pm 0.73}$ | $73.55_{\pm 0.86}$ |
| | Traffic | $43.38_{\pm 0.99}$ | $43.39_{\pm 1.01}$ | $71.21_{\pm 0.79}$ | $71.45_{\pm 0.90}$ |
| | Avg. | $44.37_{\pm 0.68}$ | $44.52_{\pm 0.68}$ | $64.11_{\pm 0.73}$ | $64.18_{\pm 0.82}$ |

