# OpenReview forum: "Repurposing Pretrained Models for Robust Out-of-domain Few-Shot Learning"
_ICLR.cc/2021/Conference — ICLR 2021 Poster_

### Official Review · AnonReviewer1 · 2020-10-25
**Nice paper and ideas**

**Rating:** 7
**Confidence:** 5

**Review:**

### Summary
This paper considers the problem of adapting a pre-trained model for few-shot learning in case there is a shift of distribution from the meta-training set. If the new tasks significantly differ from the meta-training distribution the model might need to be retrained from scratch but this is not always possible, so the authors propose to "repurpose" the model under the assumption that the support set can be used to re-calibrate the pre-trained model to the new shifted distribution.
Following two intuitions/hypotheses: 1) if the model uncertainty of specific parameters is high, then the step size should be small and 2) high uncertainty on input gradients require more adversarial training to improve robustness.
In practice, the uncertainty of the parameters is computed by using deep ensembles with perturbed MAML checkpoints rather than random initialization. High variance components are moved with lower step sizes. Moreover, If slightly perturbed models from the deep ensemble disagree on parts of the input gradient, then it means that they might be more prone to adversarial attacks, hence they need to be robustified with stronger adversarial training.

### Questions
- Q0a: maybe I'm missing the point, but am I wrong or the procedure that you propose can be also applied to any other supervised learning model, not necessarily few-shot? Let's consider you are not only observing new classes, but also new domains. Does the method apply to that situation?
- Q0b: also the use of deep ensemble and the proposed UFGSM maybe can be useful in the supervised learning scenario to improve robustness. I would suggest to try some experiments in that direction.
- Q1: from plots in figure 1 it seems that batchnorm parameters are the most uncertain. This is expected since BN capture the statistics of the activations from the meta-distribution and it is the most sensitive to the domain shift. Are you using batch-norm in a transductive setting, i.e. you don't use the running stats, but you always use the batch statistics also at test time?
    - If no, and you use running stats, then a simple baseline would be to use test time statistics, so I would add this comparison to your experiments.
    - If yes, I would also try not to update the BN scale and shift parameters in the inner loop since they seem very sensitive. Another alternative would be to have per-step running stats and per-step bn weights as proposed in Antoniou 2019.
- Q2: the improvement wrt to the SGD baseline can be more appreciated with 5 shots, rather than only 1 sample. I would have expected the opposite because of the limited number of samples. How do you interpret this?
- Q3: what happens if you try PGD instead of simply using FGSM?

### Considerations
- I think the paper is well written and motivated. The idea of repurposing a FSL model without retraining from scratch is timely and interesting. The experimental campaign is carefully performed as well as the ablation. I recommend acceptance, but I want to first to hear the author's response to my doubts.

### Bib
Antoniou et al 2019 How to train your maml

---

> ### Author Response · Authors · 2020-11-15
> **Response to AnonReviewer1 (Part 2/2)**
>
> **[Q2]**
> \
> This is a great question and we do not have a definitive answer. But we have two hypotheses why there are larger improvements in the 5-shot case than in the 1-shot case. \
> The first hypothesis is that in the 5-shot case, we can have more diverse adversarial examples than in the 1-shot case, leading to better adaptation. We have verified this on Flowers dataset for the 5-way 5-shot task, in which we used only 1, 2, 3, 4 out of 5 adversarial examples/class (denoted 5s, 10s, 15s, 20s, and 25s).
>
> ||ALL		|Top-1|		Top-40%	|
> |---|---|---|---|
> |SGD			|45.18%	|71.00%	|55.96%|
> |SGD+AT(5s)		|45.62%	|70.40%	|61.40%|
> |SGD+AT(10s)		|47.31%	|71.60%	|65.97%|
> |SGD+AT(15s)		|48.54%	|72.95%	|68.70%|
> |SGD+AT(20s)		|49.29%	|74.46%	|70.00%|
> |SGD+AT(25s)		|49.48%	|75.15%	|70.46%|
> |SGD+ALL		|53.69%	|75.30%	|72.95%|
>
> The results confirm that using more adversarial examples is beneficial, which is consistent with the improved performance gain of our method in the 5-shot (vs. 1-shot) case.
>
> The second hypothesis is that the pretrained 5-shot checkpoint has better generalization than the pretrained 1-shot checkpoint, and that the different performance gains are actually due to using different checkpoints. To verify this on 5-way classification problems, we compared the performance gain (Top-40%) of our method, when initializing from 1-shot vs. 5-shot checkpoints, evaluated on 1-shot and 5-shot classification tasks.
>
> Initializing from 1-shot checkpoint, the performance gain (SGD+ALL) - (SGD) is:
> - for 1-shot task: 45.43% - 43.69% = 1.74%
> - for 5-shot task: 62.78 % - 60.93% = 1.85%
>
> Initializing from 5-shot checkpoint, the performance gain (SGD+ALL) - (SGD) is:
> - for 1-shot task: 43.92% - 39.61% = 4.31%
> - for 5-shot task: 64.42% - 58.62% = 5.80%
>
> We conclude that the performance gain difference can be mostly explained by the checkpoint used rather than the number of shots in the evaluation task.
>
> ---
> **[Q3]**
> \
> Thank you for suggesting to apply the PGD method. We used FGSM mainly for simplicity, but it was our future plan to apply a variety of adversarial attack methods to the proposed idea.\
> As per your request, we tested PGD instead of FGSM. We have compared SGD+FGSM vs. SGD+PGD.
>
> |ALL (Avg.)	|SGD		|SGD+FGSM	|SGD+PGD	|	SGD+ALL (FGSM base)|
> |---|---|---|---|---|
> |5-way 1-shot	|36.05%	|34.36%	|35.27%		|35.94%|
> |5-way 5-shot	|40.50%	|43.67%	|42.68%		|46.74%|
>
> |Top-1 (Avg.)|	SGD	|	SGD+FGSM	|SGD+PGD|		SGD+ALL (FGSM base)
> |---|---|---|---|---|
> |5-way 1-shot	|45.36%|	46.13%	|45.70%		|46.37%
> |5-way 5-shot	|64.37%|	65.80%	|65.55%		|66.31%
>
> |Top-40% (Avg.)|SGD		|SGD+FGSM	|SGD+PGD|		SGD+ALL (FGSM base)|
> |---|---|---|---|---|
> |5-way 1-shot	|43.69%	|45.13%	|44.23%		|45.43%|
> |5-way 5-shot	|58.62%	|63.68%	|62.35%		|64.64%|
>
> Using SGD+PGD gave decent performance but did not generally outperform SGD+FGSM (except for the ALL(Avg.) metric in 5-way 1-shot). One possible reason is simply that PGD might need more hyperparameter tuning (due to lack of time we have used the ones reported in the paper [1]).
>
> More generally, those results show that other existing AT methods can also be used to improve the meta-testing stage of few-shot learning algorithms.
>
> ---
> **Reference**
> \
> [1] First-Order Adversarial Vulnerability of Neural Networks and Input Dimension

---

> > ### Comment · AnonReviewer1 · 2020-11-21
> > **Response to author's feedback**
> >
> > I'm glad that the authors found my comments useful.
> > - I was just thinking that maybe another test could be to initialize from scratch the BN beta and gamma parameters and learn them freely. We don't know for sure which is the best approach when dealing with batch normalization, especially when there is a domain-shift at test time. This could be an additional experiment for the final version of the paper.
> > - Thank you for your answer to Q2. The analysis of the number of adversarial examples and the checkpoint is definitely interesting. Nevertheless, I didn't quite get your final explanation.
> >
> > So what you are saying is that the 5-shots checkpoint shows more improvement in the OOD experiments because 5-shot tasks provide better generalization properties wrt to 1-shot tasks? Is my understanding of your answer correct? Then I guess my question is if these models are comparable or not. I think it is crucial to understand this before speculating on generalization properties and such.
> > - Did you (pre)-train the models on mini-imagenet with different hyper-parameters?
> > - How many epochs/iterations did you train the models for?
> >
> > Anyway, I still think the paper has some nice contributions so I'm raising my vote for acceptance.

---

> > > ### Comment · AnonReviewer1 · 2020-11-21
> > > **Meta-learned step size**
> > >
> > > I just wanted to add another comment.
> > > Did you try what happens if you meta-learn the stepsize? I mean something like meta-sgd.
> > >
> > > Thanks

---

> > > > ### Author Response · Authors · 2020-11-25
> > > > **Response to AnonReviewer1**
> > > >
> > > > Thank you for updating the score, and your feedback really helped us on improving our work.\
> > > > The comments have provided a good opportunity to greatly improve the quality of our work!
> > > >
> > > > ---
> > > >
> > > > **[Initialization from scratch the BN beta and gamma]**
> > > >
> > > > We didn’t try the initialization from scratch the BN beta and gamma. Thank you for your suggestion. It would be a good experiment to observe the best approach for meta-test training on domain-shift scenarios. We will do the experiments and include the results in our final version of paper.
> > > >
> > > > ---
> > > >
> > > > **[5-shot and 1-shot -  Is my understanding of your answer correct?]**
> > > >
> > > > Not quite, as we think we were not clear enough. We wish to correct the phrasing of our answer for 5-shot vs. 1-shot performance gain (please replace “better generalization” with “better performance gain”). We are merely observing that the performance gain is empirically dominated by the choice of checkpoint. As the reviewer noted, it is tricky and outside of the scope of this paper to compare generalization properties of the different models.
> > > > - Did you (pre)-train the models on mini-imagenet with different hyper-parameters? \
> > > > = We trained the models on a mini-imagenet with the same hyper-parameters.
> > > > - How many epochs/iterations did you train the models for?\
> > > > = Unless specified otherwise, we selected the checkpoints using early-stopping, based on mini-ImageNet validation performance, with highest accuracy at the 57,200 (5-way 1-shot) and 45,600 iterations (5-way 5-shot) -- see Appendix A.4. \
> > > > We have also compared initializing from early-stopped 5-shot and overfitted 5-shot checkpoints (at 150,000 iterations). Similarly to the early-stopped models, we observed large performance gains using overfitted 5-shot. See detailed results in Discussion 4.2 (Best checkpoint vs. Overfitted checkpoint.) and Appendix A.8.
> > > >
> > > > Interestingly, note that several methods from the few-shot literature tend to yield bigger performance gains for the 5-shot case than the 1-shot case. We cite below results from different papers comparing the performance of some standard few shot-learning baseline with their proposed method. The below tables report accuracies on miniImageNet for 5-shot vs. 1-shot learning and the performance gain.
> > > >
> > > > Thank you for your fruitful feedback and insightful suggestion.
> > > >
> > > > ---
> > > >
> > > > **[Meta-learned stepsize]**
> > > >
> > > > This is an interesting experiment for few-shot learning in general. However this is not possible in our specific setting where it is assumed that the meta-training set is not accessible.
> > > >
> > > > ---
> > > >
> > > > **<Accuracies on miniImageNet for 5-shot vs. 1-shot learning and the performance gain>**
> > > >
> > > > |Reptile [1] | 5-way 1-shot | 5-way 5-shot |
> > > > |---|---|---|
> > > > |MAML+Transduction | 48.7 | 63.11 |
> > > > |Reptile + Transduction |49.97 |65.99 |
> > > > |Gain wrt MAML |1.27 |**2.88** |
> > > >
> > > > | MAML++ [2] | 5-way 1-shot | 5-way 5-shot |
> > > > |---|---|---|
> > > > |MAML (local reproduction) |48.28 |64.39 |
> > > > |MAML++ |52.15 |68.32 |
> > > > |Gain wrt MAML |3.87 |**3.93**|
> > > >
> > > > |Prototypical Networks [3] |5-way 1-shot |5-way 5-shot |
> > > > |---|---|---|
> > > > |MAML |48.7 |63.15 |
> > > > |Prototypical Networks |49.42 |68.2 |
> > > > |Gain wrt MAML |0.72 |**5.05**|
> > > >
> > > > |Meta-SGD [4]|5-way 1-shot|5-way 5-shot|20-way 1-shot|20-way 5-shot|
> > > > |---|---|---|---|---|
> > > > |MAML|48.7|63.11|16.49|19.29|
> > > > |Meta-SGD|50.47|64.03|17.56|28.92|
> > > > |Gain wrt MAML|1.77|0.92|1.07|**9.63**|
> > > >
> > > > |Meta-Learner LSTM [5]|5-way 1-shot|5-way 5-shot|
> > > > |---|---|---|
> > > > |Matching Network|43.56|55.31|
> > > > |Meta-Learner LSTM|43.44|60.6|
> > > > |Gain wrt Matching Network|-0.12|**5.29**|
> > > >
> > > > ---
> > > >
> > > > **References**
> > > >
> > > > [1] Alex Nichol, Joshua Achiam, and John Schulman. On first-order meta-learning algorithms. arXiv preprint arXiv:1803.02999, 2018.\
> > > > [2] Antreas Antoniou, Harrison Edwards, and Amos Storkey. How to train your MAML. In Proceedings of the International Conference on Learning Representations, 2019.\
> > > > [3]  Jake Snell, Kevin Swersky, and Richard Zemel. Prototypical networks for few-shot learning. In Advances in Neural Information Processing Systems, pp. 4077–4087, 2017.\
> > > > [4] Sachin Ravi and Hugo Larochelle. Optimization as a model for few-shot learning. In Proceedings of the International Conference on Learning Representations, 2017.\
> > > > [5] Zhenguo Li, Fengwei Zhou, Fei Chen, and Hang Li. Meta-sgd: Learning to learn quickly for few-shot learning. arXiv preprint arXiv:1707.09835, 2017.

---

> ### Author Response · Authors · 2020-11-15
> **Response to AnonReviewer1 (Part 1/2)**
>
> Thank you for such a thorough review! We are grateful for your insightful comments and glad that you like many aspects of the paper. Some of your fruitful suggestions even helped improve the results.
> \
> \
> **[Q0]**
> \
> Thank you for suggesting to apply our method to supervised learning.\
> Indeed, our method can be integrated into any supervised learning framework, as long as ensemble training and adversarial training are possible.\
> This work focuses only on the few-shot learning scenario, but we expect our method to be a helpful tool for supervised learning and modeling model uncertainty, which could be explored in future work.
>
> ---
> **[Q1]**
>
> The reviewer is correct in noting that BN parameters have the most uncertainty (Fig 1).\
> Note that we have used the transductive setting throughout the paper (using minibatch statistics of the support set and query set during meta-testing).
>
> Following the reviewer’s suggestion, we ran new experiments in which we froze the BN parameters. We compared SGD+ALL (w/ freezing BN) and SGD+ALL (w/o freezing BN).\
> (Please understand that we tested one seed due to lack of time and computation resources).
>
> <5-way 1-shot results on SGD+ALL (ALL metric)>
>
> |	|w/ freezing BN	|	w/o freezing BN|
> |---|---|---|
> |Bird	|	30.99%	|	30.03%|
> |Flower	|	44.54%	|	41.53%|
> |Mini	|	37.40%	|	36.46%|
> |Traffic|	37.34%	|	35.73%|
> |Average|	37.57%	|	35.94%|
>
> <5-way 5-shot results on SGD+ALL (ALL metric)>
>
> |		|w/ freezing BN		|w/o freezing BN|
> |---|---|---|
> |Bird		|41.13%		|38.73%|
> |Flower		|60.04%		|53.62%|
> |Mini		|45.64%		|43.68%|
> |Traffic	|54.62%		|50.93%|
> |Average	|50.36%		|46.74%|
>
>
> It turns out SGD+ALL (w/ freezing BN) outperformed SGD+ALL (w/o  freezing BN) on the ALL metric, with most improvement in the higher stepsize range (>0.1).
>
> <5-way 1-shot results on SGD (ALL metric)>
>
> |		|w/ freezing BN		|w/o freezing BN|
> |---|---|---|
> |Bird		|30.37%		|30.49%|
> |Flower		|40.84%		|41.21%|
> |Mini		|36.83%		|37.37%|
> |Traffic	|35.60%		|35.11%|
> |Average	|35.91%		|36.05%|
>
> <5-way 5-shot results on SGD (ALL metric)>
>
> |		|w/ freezing BN|w/o freezing BN|
> |---|---|---|
> |Bird		|34.78%		|33.25%|
> |Flower		|49.76%		|44.59%|
> |Mini		|39.59%		|38.93%|
> |Traffic	|48.00%		|45.25%|
> |Average	|43.03%		|40.50%|
>
> Also in the 5-way 5-shot case, SGD (w/ freezing BN) outperformed SGD (w/o freezing BN) on the ALL metric, with most improvement in the higher stepsize range. But in the 5-way 1-shot case, there was no performance gain, because 5-way 1-shot didn’t show poor accuracy at high stepsize (>0.1).
>
> This is consistent with our intuition that updating high-uncertainty layers (such as BN) with large stepsizes can be hurtful.
>
> We are very grateful for your feedback and have used it to improve our paper.
> The new experiments can be found in Section 4.2 (Detailed results are in Appendix A.9).

---

### Official Review · AnonReviewer3 · 2020-10-26
**A good first attempt to solve a new practical problem. However, why can’t the problem be solved by existing domain transfer or domain adaptation methods?**

**Rating:** 6
**Confidence:** 4

**Review:**

The paper tries to solve a novel practical problem: adapting pretrained meta-learning checkpoints to out-of-domain test tasks. It is an important problem in industry since there are some industrial tasks, especially meta-learning tasks, that have very limited data. Therefore, the meta-training process which requires a relatively large data set is not practical.

Towards this end, the paper proposes to run ensembling fine-tuning with the support set of the test task, and use the weight variances from this process to guide the training.  The paper also proposes to add task adversarial examples to the training set to help the meta fine-tuning process. Experimental results seem promising.

One main concern I have in mind is, existing domain adaptation or domain transfer methods are trying to solve very similar problems, i.e., the model is pre-trained with large data set from other domains and you need to adapt it to the new domain. What are the differences between the current problem and domain transfer/adaptation problems? Why can’t we apply existing domain adaptation methods to solve the current problem?

---

> ### Author Response · Authors · 2020-11-15
> **Response to AnonReviewer3**
>
> Thank you for your review and appreciating the industrial motivation for our novel problem!
>
>
> We’d like to bring an important clarification about your fair question about domain adaptation.
>
>
> Most domain adaptation methods, such as those based on GANs, require having access to the source (meta-training) dataset. This paper was motivated by a specific setting in which we *only* have access to the pretrained checkpoint, but not to the meta-training set (say, due to confidentiality or privacy issues).
>
>
> In addition, unlabeled target domain samples required for unsupervised domain adaptation methods are often not easy to access. We will modify shortly the paper to discuss more the related work of domain adaptation.
>
> However, our proposed ideas (perturbation-based ensemble model training and uncertainty estimation, USA and UFGSM) can also be applied to other problems, such as supervised learning, as suggested by Reviewer 1. In fact, it is theoretically possible to combine our method with domain adaptation techniques, as long as those are compatible with ensemble-model training and adversarial-example training.
>
>
> We agree that it is interesting future work to extend our method to transfer learning (See last sentence of conclusion).

---

> > ### Comment · AnonReviewer3 · 2020-11-17
> > **Thanks for your reply!**
> >
> > The authors have addressed my concerns and I have changed my rating accordingly.

---

> > > ### Author Response · Authors · 2020-11-25
> > > **Response to AnonReviewer3**
> > >
> > > Thank you for updating the score, and your feedback really helped us on improving our work.

---

### Official Review · AnonReviewer4 · 2020-10-27
**Official Blind Review #4**

**Rating:** 5
**Confidence:** 4

**Review:**

[Summary]
MAML has two stages: (1) In the meta-training stage, given various tasks (but each say only has a few labeled data), we want to arrive at a representation where it can quickly adapt to any test task later. Let us denote this as $\theta_0$ (2) In the meta-testing stage, we go from $\theta_0$, given the few-shot test data, via an SGD process (typical scenario), to get a final model that has good performance on the test data of the test task.

In MAML, we assume that tasks are (i.i.d.) sampled from a distribution of tasks P(T). This paper considers a setting (as I understand) where the test task is out-of-distribution of P(T). In this setting, the authors argued further that one cannot run the meta training, so the focus of the paper is to improve stage (2) (gradient update process at the test time).

The proposed method contains two main parts: (A) We use an ensemble method to help understand uncertainty of the network, and finally use different step sizes for different layers for updating the network parameters at test adaptation phase, (B) adversarial training as data augmentation to help the test-time adaptation process. Experiments show that this method can bring some performance improvements

[Assessment]
1. I am not sure I entirely accept the authors' statement about testing tasks being out-of-distribution compared to the training tasks. In the original MAML setting, the test tasks can be entirely different from the training tasks (for example, we pick test tasks from different classes). Further, it has always been the case in MAML, that one can store a checkpoint $\theta_0$ trained from some tasks, and  when a new task (maybe entirely different) comes, we start from $\theta_0$ to adapt. This renders the motivations of this work less meaningful. And indeed, if we look at the performance numbers, the improvement is somewhat marginal.

2. More specifically, at the very least, I think the precision of the paper can be improved. If MAML's purpose is already to be able to adapt to new, unseen, tasks, what is the precise meaning of out-of-distribution tasks here? (for example, one interpretation of this, stated in a formal way, is what I described above as saying that the new task is out-of-distribution of P(T))

3. I do like the use of ensemble and also adversarial examples, and intuitively they can improve the test-time adaptation. However, the arguments for why they are useful are quite ad-hoc (to a degree that I feel superficial, no offense). Can we better justify their usage?

4. What is the computational overhead in order to gain the additional accuracy? The use of ensemble together with adversarial training seems to bring a lot of computation cost, which seems not a good deal in view of the accuracy gain.

---

> ### Author Response · Authors · 2020-11-15
> **We would like to thank you for the thorough review of the manuscript and for insightful questions.  We believe that our important clarifications below should address your raised concerns and hope that your assessment could be updated.**
>
> **[Q1 & Q2]**
> \
> *<Meaning of “Out-of-distribution Task”>*
> \
> In our study the precise meaning of out-of-distribution is a cross-domain task, in our case we repurpose a model meta-trained on miniImageNet for meta-testing on other datasets like Bird, Flower and Traffic signs (We have also tested on miniImageNet for completeness). This reflects the terminology used in the out-of-distribution literature [1,2,3], which mention “The key observation is that the distributions of the image features extracted from tasks in the unseen domains are significantly different from those in the seen domains” [3]. The experimental setting of the papers [1][2] refers to the case where the training dataset and test dataset are different data sets(domain). At training time they trained the model on CIFAR-10 or CIFAR-100. At test time, they use test images from CIFAR-10 (CIFAR-100) dataset as in-distribution examples, and TinyImageNet and LSUN as out-of-distribution examples.
>
>
> *<MAML can be applied to our setting>*
> \
> Our motivation assumes a new few-shot learning setup where additional restrictions exist :
> - No access to meta-train dataset
> - Solving cross-domain (“out-of-distribution”) few-shot classification tasks
>
> The reviewer is correct that MAML can be applied to our setting, which is why we have taken MAML as a baseline (denoted SGD).
>
>
> *<Limited Improvement>*
> \
> The 5-way 1-shot did not show a significant performance improvement, but performance improvements had been observed in the overall metrics. In addition we improved 1.52% accuracy on ALL metric by freezing the most uncertain layer (BN) during this review process. In 5-way 5-shot case, performance was improved 6.08%, 1.58% and 5.84% respectively in All, Top-1 and Top-40%. In particular, when the pretrained model showed unstable performance, such as overfitting, it shows much greater performance improvement (7.48%, 2.2% and 7.28% in All, Top-1, Top-40%, respectively).
> ---
>
> **[Q3]**
> \
> The only strong evidence is empirical. It is difficult to come up with a more precise theory of why ensemble & adversarial training help. But Xie et al. (2020)[4] reported improvement using generated examples for adversarial attack on the large scale training. Tramer et al. (2018)[5] introduced Ensemble Adversarial Training, a technique that augments training data with perturbations transferred from other models. Ensemble Adversarial Training yields models with stronger robustness to blackbox attacks on ImageNet. From this result, we can see that the adversarial example generated through static pre-trained models can provide more diverse features to the model. Of course our ensemble models are generated from a single pre-trained model by gaussian perturbation. Perturbed models do not have enough different characteristics but as each ensemble model is trained, it can have different characteristics. Therefore generated adversarial examples from each ensemble model are different.  Also we can know intuitively that it is advantageous to use a lot of diverse data with different characteristics to improve model training performance. The generated adversarial examples during ensemble training have subtle different characteristics because they are generated from different models. Through this related literature and our intuition, we use the generated adversarial examples during ensemble model training for improving model performance. Our approach can be regarded as a form of data augmentation.
> ---
>
> **[Q4]**
> \
> In our experience, an important quality in the industry is to have out-of-the-box and easily tunable methods. Although our method does not necessarily improve the Top-1 accuracy a lot, the All and Top-40 metrics show improved robustness to the choice of step sizes.
> There is indeed a computational overhead proportional to the size of the ensemble. However, 1) since it is few shot learning, the support set is so small that the computational overhead is negligible, and 2) our method incurs no overhead during inference (meta-testing query set), since prediction on the query set is equivalent to MAML inference (only with modified parameters),
>
>  ---
> References\
> [1] Shalev, Gabi, Yossi Adi, and Joseph Keshet. "Out-of-distribution detection using multiple semantic label representations." Advances in Neural Information Processing Systems. 2018.\
> [2] Liang, Shiyu, Yixuan Li, and R. Srikant. "Enhancing The Reliability of Out-of-distribution Image Detection in Neural Networks." International Conference on Learning Representations. 2018.\
> [3] Tseng, Hung-Yu, et al. "Cross-Domain Few-Shot Classification via Learned Feature-Wise Transformation." International Conference on Learning Representations. 2020.\
> [4] Xie, Cihang, et al. "Adversarial examples improve image recognition." Proceedings of the IEEE/CVF Conference on Computer Vision and Pattern Recognition. 2020.\
> [5] Tramèr, Florian, et al. "Ensemble Adversarial Training: Attacks and Defenses." International Conference on Learning Representations. 2018.

---

> > ### Comment · AnonReviewer4 · 2020-11-21
> > **Thanks for the detailed responses**
> >
> > They addressed some of my concerns and thus I raised my score. A reason that I am still reluctant to accept is that the techniques in the paper feel ad-hoc (as I mentioned in my review) and it is hard to justify the principles, as the authors have agreed (for example, using different step sizes at different layers).

---

> > > ### Author Response · Authors · 2020-11-25
> > > **Response to AnonReviewer4**
> > >
> > >
> > > Thank you for updating the score, and your feedback really helped us on improving our work.
> > >
> > >
> > > *<< A reason that I am still reluctant to accept is that the techniques in the paper feel ad-hoc (as I mentioned in my review) and it is hard to justify the principles, as the authors have agreed (for example, using different step sizes at different layers).>>*
> > >
> > >
> > > We’d like to clarify that we do not believe our principles to be ad hoc (there is a difference between not having a formal mathematical statement and having no argument whatsoever). In addition to the justification for ensemble & adversarial training mentioned in the previous rebuttal, the use of different step sizes at different layers has even better justification, as we described in the paper, the motivation for using layer-wise stepsize is because features from the same layer tend to have the same level of abstraction[1] (Please see footnote at page 5). Also  there are some literatures related with layer-wise stepsize that can improve training speed and generalization performance[2][3][4]. We would like to clarify that our motivation and related literatures are rational reasons to use layer-wise stepsize in our method.
> > >
> > > Our study contributes not only to new techniques for improving performance in the meta-testing phase but also to the definition of the novel problem in few-shot learning that can happen in real-world. Also Our results motivate the use of adversarial learning as a data augmentation scheme for improving few-shot generalization.
> > >
> > > While we are not providing a formal theoretical justification for the method, we supported our hypothesis empirically. Our empirical observation of improving over the default meta-testing procedure of MAML motivates further research on alternative ways to leverage published model checkpoints.
> > >
> > > ---
> > >
> > > **References**
> > >
> > > [1] Matthew D Zeiler and Rob Fergus. Visualizing and understanding convolutional networks. In ECCV, 2014.\
> > > [2] You, Yang, Igor Gitman, and Boris Ginsburg. "Scaling sgd batch size to 32k for imagenet training." arXiv preprint arXiv:1708.03888 6 (2017).\
> > > [3] B. Singh, S. De, Y. Zhang, T. Goldstein, and G. "Taylor. Layer-specific adaptive learning rates for deep networks". ICMLA, 2015.\
> > > [4] Belilovsky, Eugene, Michael Eickenberg, and Edouard Oyallon. "Greedy layerwise learning can scale to imagenet." PMLR, 2019.

---

### Official Review · AnonReviewer2 · 2020-10-28
**submission 831 review**

**Rating:** 5
**Confidence:** 2

**Review:**

The paper proposes to reutilize pretrained MAML checkpoints for out-of-domain few-shot learning, combining with uncertainty-based adversarial training and deep ensembles.

Pros:

1. The idea of combining meta-learning, uncertainty learning and adversarial training is well-structured. In particular, the related work part provides a clear introduction of background work.

2. It is quite novel to leverage adversarial learning as data augmentation for meta-testing in MAML.
3. The paper provides extensive and convincing experiment results over evaluating the proposed model’s robustness to the choice of base stepsizes.

Cons:

1. It would be better if an optimization equation is provided, especially if there are generated adversarial examples.

2. For the ablation study, the authors mention that the best absolute performance (Top-1) is always obtained through some use of adversarial training. Actually, it would be more convincing if they can discuss more choices of \lambda_{AT} and \lambda_{AUG} and \lambda_{a} to present the sensitivity analysis of the hyper-parameters.

---

> ### Author Response · Authors · 2020-11-15
> **Response to AnonReviewer2**
>
> Thank you for taking the time to review this paper. We provide important clarifications to the two "cons" below:
> \
> \
> **Con 1.**
> \
> **As specified on line 15 of Algorithm 1,** we trained the model by minimizing the summation of the following three losses.
> - Cross-entropy loss on support set
> - Cross-entropy loss on generated adversarial examples from support set
> - Cross-entropy loss on generated adversarial examples during ensemble training and generated by UFGSM.
>
> Let us know if it is still unclear to you.
>
> ---
>
> **Con 2.**
> \
> We use the lambda notation to easily describe various combinations of our method.
> In our algorithm the lambdas are binary (see “Require” section of Alg 1).
> Each lambda is set to a fixed value in our full method. (see third paragraph of section 3.2).
> We have studied various combinations of binary of lambdas in the ablation studies.
> More combinations are theoretically possible but beyond the scope of the paper.
>
> Thank you for your suggestion. Also your suggestions can help to understand how each loss value can affect the model training.

---

### Author Response · Authors · 2020-11-25
**Summary of the updates in the revision.**

**Summary of the updates in the revision.**

We thank all reviewers for their constructive comments. During the rebuttal period we revised the paper to faithfully reflect the comments from all the reviewers.
We have made the following modifications to the paper:

1. We included domain adaptation in Related Works suggested by R3.
2. We included in the experimental results for freezing BN suggested by R1 (Section 4.2 and detailed results are in Appendix A.9).
	- We updated the Table 10 (results of freezing BN layers experiment) from the previous version (14 Nov).
	- Previous version (14 Nov) tested only one seed. Current version (23 Nov) was tested on 6 different seeds. Note that using more seeds did not change our previous observations.
3. We described our motivation and problem setting more clearly (Main contribution part in Introduction).

---

### Decision · Program_Chairs · 2021-01-07
**Final Decision**

**Decision:**

Accept (Poster)

**Comment:**

This paper considers a new and practical setting of meta-learning for out-of-domain task adaptation where a pretrained model exists but the original meta-training data is not available. The authors incorporate several ideas including deep ensembles, adversarial training and uncertainty-based step sizes, and achieve competitive performance under this particular setting.

The combination of various methods appears complicated, but the authors provide detailed ablation study to show the effectiveness of each component empirically. During rebuttal and discussion, they addressed many of the concerns from the reviewers. As pointed out by a reviewer, their proposed method would have a value in the domain adaptation area beyond meta-learning.

The remaining concern is on the somewhat ad-hoc combination of multiple methods and lack of a clear single solution for addressing the OOD few-shot learning problem. Nonetheless, the proposed methods show a convincing empirical improvement on the vanilla MAML baseline in the experiments.